# Bayesian parameter inference in hydrological modelling using a Hamiltonian Monte Carlo approach with a stochastic rain model

Simone Ulzega[1] and Carlo Albert[2]

[1]Institute of Computational Life Sciences, Zurich University of Applied Sciences, Wädenswil, Switzerland
[2]Swiss Federal Institute of Aquatic Science and Technology, Dübendorf, Switzerland
**Correspondence:** Simone Ulzega (simone.ulzega@zhaw.ch)

**Abstract.**

Stochastic models in hydrology are very useful and widespread tools for making reliable probabilistic predictions. However, such models are only accurate at making predictions if model parameters are first of all calibrated to measured data in a consistent framework such as the Bayesian one, in which knowledge about model parameters is described through probability distributions. Unfortunately, Bayesian parameter calibration, a.k.a. inference, with stochastic models, is often a computationally intractable problem with traditional inference algorithms, such as the Metropolis algorithm, due to the expensive likelihood functions. Therefore, the prohibitive computational cost is often overcome by employing over-simplified error models, which lead to biased parameter estimates and unreliable predictions. However, thanks to recent advancements in algorithms and computing power, full-fledged Bayesian inference with stochastic models is no longer off-limits for hydrological applications. Our goal in this work is to demonstrate that a computationally efficient Hamiltonian Monte Carlo algorithm with a time-scale separation makes Bayesian parameter inference with stochastic models feasible. Hydrology can potentially take great advantages from this powerful data-driven inference method as a sound calibration of model parameters is essential for making robust probabilistic predictions, which can certainly be useful in planning and policy making. We demonstrate the Hamiltonian Monte Carlo approach by detailing a case study from urban hydrology. Discussing specific hydrological models or systems is outside the scope of our present work and will be the focus of further studies.

## 1 Introduction

A fundamental and highly non-trivial question in many applied sciences is how to make reliable predictions about the dynamics of a complex system. In hydrological modelling in particular the ability of predicting extreme events like floods is obviously of paramount importance. Conceptual rainfall-runoff models that incorporate only a few state variables and a few system parameters often represent a very practical and efficient solution for making probabilistic predictions. The basic idea is to describe slow processes occurring at our observation scale by phenomenological differential equations and include all other processes as noise. Incorporating the noise in the model, where it arises, naturally leads to stochastic differential equation (SDE) models. Model parameters then need to be calibrated on observed data, usually provided in the form of noisy time series. The goal of the calibration process is to determine the parameters that allow the model to reproduce the observed

data and quantify their uncertainties, expressed as probability distributions. For this purpose, Bayesian statistics is a consistent framework where our knowledge about model parameters is described by probability distributions and learning as a data-driven updating process of prior beliefs. Bayesian inference methods bear the great advantage over traditional optimization algorithms of providing an uncertainty estimation for the calibrated parameters in the form of a probability distribution. The knowledge of such uncertainty is important for making probabilistic predictions, which can be in turn a useful tool for decision makers.

Hydrology could potentially take great advantages from more realistic stochastic models and a fast and reliable method for their calibration. However, Bayesian inference turns out to be computationally very expensive for non-trivial stochastic models.

Uncertainty in rainfall-runoff hydrological modelling arises mostly from input errors associated with an inaccurate estimation of the integrated rainfall over a catchment (Kavetski et al., 2006). These input errors, typically due to a combination of heterogeneous rainfall, sparse rain-gauge measurements and insufficient temporal resolution (McMillan et al., 2011; Renard

et al., 2011; Ochoa-Rodriguez et al., 2015), can seriously deteriorate the quality of model calibration results for heavily input-driven hydrological systems (Bárdossy and Das, 2008). A variety of stochastic weather generators (Rodriguez-Iturbe et al., 1987; Cowpertwait et al., 1996; Deidda et al., 1999; Paschalis et al., 2013; Langousis and Kaleris, 2014) have been proposed to simulate precipitation with its uncertainty. Although such weather generators can provide uncertain inputs to rainfall-runoff models, and therefore reproduce the effect of rainfall errors on runoff predictive uncertainties, such input uncertainties have

been largely neglected in studies focusing on model parameter inference (Sikorska et al., 2012), probably because of the computational difficulty of including them in a likelihood function (Honti et al., 2013). Input uncertainties should be included directly in the input as a stochastic contribution, then transported through the model and thereby naturally incorporated in the likelihood function describing the probability distribution of observations given model parameters. In the Bayesian framework, the sought posterior probability distribution for the model parameters is proportional to the product of the likelihood function and

the prior probability distribution describing our prior knowledge about model parameters. However, it is still common practice in hydrology, as well as in other applied disciplines, to consider an over-simplified error model based on likelihood functions defined as uncorrelated normal distributions centred on the outputs of a deterministic model (Yang et al., 2008; Reichert and Schuwirth, 2012; Sikorska et al., 2012). This inevitably leads to biased parameters and unreliable predictions (Renard et al., 2011; Honti et al., 2013; Del Giudice et al., 2015). Although so-called rainfall multipliers can mitigate this problem (Kavetski

et al., 2006; Sun and Bertrand-Krajewski, 2013), they fail in assessing input uncertainties when a rainfall event is not detected by the available rain-gauges (Kavetski et al., 2006; Renard et al., 2011).

Del Giudice et al. (2016) proposed a method based on an input uncertainty model describing the rainfall as a continuous stochastic process. The method is called SIP (acronym for Stochastic Input Process). The idea of describing selected model parameters or inputs as stochastic time-dependent processes in order to take intrinsic uncertainties realistically into account in

hydrological modelling has gained momentum in recent years (Tomassini et al., 2009; Reichert and Mieleitner, 2009; Reichert et al., 2021; Bacci et al., 2022). The SIP technique uses i) possibly inaccurate rain-gauge precipitation data, ii) runoff data from a flowmeter at the catchment outlet, iii) a hydrological runoff model, iv) a rainfall model in the form of a transformed stochastic Ornstein–Uhlenbeck (OU) process, v) models for rainfall and runoff observation errors, and vi) prior distributions, to infer in a Bayesian manner both the marginal posterior distributions for the parameters of interest and a "true" spatially-integrated

average rainfall over the catchment. The SIP method uses the catchment as an additional rain-gauge to gather information about a catchment-averaged real rain, which is inferred from both prior knowledge and observations of rainfall and runoff. Both the hydrological model describing the catchment dynamics and runoff observations are assumed to be accurate in comparison to the rainfall data. The inferred rainfall pattern can then be used to calibrate the model and significantly reduce the bias in the estimated parameters, despite the inaccuracy of the rainfall data. However, as described in detail in Del Giudice et al. (2016), the likelihood function turns out to be a high-dimensional discretized version of an infinite-dimensional path-integral that makes this approach computationally demanding.

In Albert et al. (2016) we presented an efficient Hamiltonian Monte Carlo (HMC) based algorithm for the calibration of SDE models on noisy time series. The proposed HMC method, combining molecular dynamics principles with the well-known Metropolis algorithm, relies on the reinterpretation of the Bayesian posterior probability distribution as the partition function of a statistical mechanics system. This interpretation reduces the parameter inference problem to the task of simulating the dynamics of a fictitious statistical mechanics system, which can be solved in a computationally efficient way. The dynamics of the statistical mechanics system may occur on very different time scales, which can be exploited by a multiple time-scale integration approach. In Albert et al. (2016), the method was demonstrated using a simple, albeit general, rainfall-runoff toy model, and we claimed that the HMC algorithm, combined with the multiple time-scale integration, would be applicable to a wide range of inference problems, making many SDE models amenable to a consistent Bayesian parameter inference.

Here we combine the HMC and SIP methods to perform Bayesian inference with a stochastic input model. We show that the HMC method can be extended from the toy model and the smooth synthetic data used in Albert et al. (2016) to a real-world hydrological case study with real noisy rainfall and runoff time series, albeit at the cost of a somewhat substantial analytical effort. We intentionally use sparse and inaccurate precipitation data, provided by a rain-gauge located far away from the catchment area, to demonstrate that a "true" average rainfall pattern can be reconstructed from the corresponding runoff data, and we compare the inferred rain to the more accurate observations obtained from rain-gauges within or very close to the catchment.

The HMC method bears valuable advantages with regard to both generality and efficiency. Indeed, it is by no means limited to an OU process, unlike the original SIP approach of Del Giudice et al. (2016), which strictly requires a linear stochastic process as a rain generator. Although in this study we also opt for an OU process for the sake of simplicity, it should be clear that such process could be arbitrarily replaced by any other stochastic process, thus giving the method significantly more flexibility in reproducing the statistical properties of real rainfall. The HMC method is also not at all limited to urban hydrology, and could be applied for instance in natural catchment hydrology as well. The specific case study presented here was chosen to put emphasis on input uncertainties, which often represent the biggest source of uncertainty in hydrological modelling in general. More information on possible hydrological applications can be found in Del Giudice et al. (2016). Moreover, the HMC algorithm allows us to sample from the posterior probability density both model parameters and a true averaged rainfall pattern that are simultaneously compatible with data, models and prior distributions. This yields very high acceptance rates even in the context of expensive high-dimensional problems, with great benefits in terms of performance and efficiency of the algorithm. Another interesting method from the family of particle filters to tackle high-dimensional inference problems

for stochastic model calibration is Particle Markov Chain Monte Carlo (PMCMC) (Andrieu et al., 2010), which combines piece-wise forward simulations of the stochastic model with data-based importance sampling. Like HMC, PMCMC methods can be applied to any stochastic process, including (unlike HMC) processes with discrete states (e.g., numbers of organisms in ecological models). A generally low implementation effort could be an additional argument in favor of PMCMC algorithms, which do not require the differentiation of the posterior distribution like HMC methods. However, PMCMC methods may suffer significantly in terms of efficiency when compared to an HMC-based approach. A detailed comparison, based on the same case study presented here, of different methods for Bayesian inference with stochastic models, can be found in Bacci et al. (2023). The HMC approach is very general and suitable for a broad range of applications requiring Bayesian inference with stochastic models. It should be noted that the method presented here is meant for offline-calibration of stochastic models, not for real-time updating, which might be needed in a model-based control setting. For the latter problem, filtering algorithms might be more appropriate, but we do not discuss this topic here. Moreover, HMC is inherently a MCMC method and it is therefore embarrassingly parallelizable by breaking it up into an arbitrary number of independent Markov chains. This makes HMC very well-suited for applications in the big-data regime or with expensive models.

## 2   Bayesian inference with a stochastic rain model

The SIP method of Del Giudice et al. (2016) describes the rain input to a hydrological system based on an unobserved and continuous stochastic process, the rainfall potential $\xi(t)$. The latter should not be interpreted as a potential in the physical sense, but rather as a rain generator based on a latent, i.e., "potential", stochastic process that can be transformed into real precipitation $P$ by a suitable empirical transformation $P(t) = r(\xi(t))$, which will be addressed below in Section 2.1.

The inference process allows us to learn from noisy rainfall and runoff time series, $\boldsymbol{P}_{\mathrm{obs}}$ and $\boldsymbol{Q}_{\mathrm{obs}}$, respectively, the parameters $\boldsymbol{\theta}$ of the hydrological system, the uncertainties $\sigma_\xi$ and $\sigma_z$ of both rainfall and runoff observation models, respectively, and the unobserved true rainfall over the catchment expressed as a discretized rainfall potential $\boldsymbol{\xi} = \{\xi_i\}$, to be interpreted as an evaluation of the stochastic process $\xi(t)$ at a discrete set of time points $t_i$, with $i = 1, ..., N$.

For this purpose, we subdivide each interval between consecutive rain observations into $j_P$ bins, yielding a total of $n_P j_P + 1 = N$ discretization points, where $n_P + 1$ is the number of rain observations, that is, the length of the precipitation time-series $\boldsymbol{P}_{\mathrm{obs}}$. Analogously, the system output dimension is discretized by partitioning the intervals between consecutive runoff observations into $j_Q$ sub-intervals, such that the total number of discretization points is $n_Q j_Q + 1 = N$, where $n_Q + 1$ is the number of data points in $\boldsymbol{Q}_{\mathrm{obs}}$. The number of discretization points is thus the same $(N)$ in both rainfall and runoff dimensions, and it defines the discretization time step $dt = T/(N-1)$, where $T$ is the total time interval covered by observations. We will provide numerical values for all these constants later in Section 4. Here, suffice it to say that the only important requirement of the method is that the total number $N$ of discretization points is large enough compared to the number of measurement points, $n_P$ and $n_Q$, in order to accurately probe the fine dynamics occurring on short time scales between observations. Other features, such as for instance having the same number $N$ of discretization points for rainfall and runoffs, are just arbitrary choices to ease the practical implementation of the method, which could be removed without altering the results and conclusions of this

work. The same holds if the observations were irregularly spaced in time, in which case one could use observation windows with more/less intermediate discretization points. Note that a large number of discretization points $N$, with a fixed number of observation points, does only moderately increase the computational effort of the algorithm, since the part of the Hamiltonian dynamics that scales with $N$ can be calculated analytically, as shown below.

The inference goal is to sample both parameter combinations $(\boldsymbol{\theta}, \sigma_\xi, \sigma_z)$ and realizations $\boldsymbol{\xi}$ of the stochastic process $\xi(t)$ from a posterior distribution obeying Bayes' equation, which reads in its discretized form as,

$$f(\boldsymbol{\xi}, \boldsymbol{\theta}, \sigma_\xi, \sigma_z \mid \boldsymbol{P}_{\mathrm{obs}}, \boldsymbol{Q}_{\mathrm{obs}}) \propto f(\boldsymbol{Q}_{\mathrm{obs}} \mid \boldsymbol{\xi}, \boldsymbol{\theta}, \sigma_z) \, f(\boldsymbol{P}_{\mathrm{obs}} \mid \boldsymbol{\xi}, \sigma_\xi) \, f(\boldsymbol{\xi}) \, f(\boldsymbol{\theta}, \sigma_\xi, \sigma_z) \,, \tag{1}$$

where $f(\boldsymbol{\theta}, \sigma_\xi, \sigma_z)$ is the joint prior distribution for the model parameters, and $f(\boldsymbol{Q}_{\mathrm{obs}} \mid \boldsymbol{\xi}, \boldsymbol{\theta}, \sigma_z) \, f(\boldsymbol{P}_{\mathrm{obs}} \mid \boldsymbol{\xi}, \sigma_\xi)$ is the *likelihood* expressing the probability of observed data $(\boldsymbol{P}_{\mathrm{obs}}, \boldsymbol{Q}_{\mathrm{obs}})$ given model parameters $(\boldsymbol{\theta}, \sigma_\xi, \sigma_y)$ and a system realization $\boldsymbol{\xi}$, weighted by the prior probability density $f(\boldsymbol{\xi})$.

The stochastic process realization $\boldsymbol{\xi}$ in Eq. 1 is a time-series of length $N \gg 1$. The high dimensionality of the problem renders the likelihood computationally expensive, thus making the inference problem intractable with traditional Bayesian inference algorithms, such as random walk Metropolis algorithms, which require a large number of likelihood evaluations. On the other hand, when the inference target is only the posterior distribution for the model parameters $(\boldsymbol{\theta}, \sigma_\xi, \sigma_z)$ and model simulations are fast, one may resort to Approximate Bayesian Computation (ABC) algorithms (e.g., Albert et al. (2015)), which approximate the parameters posterior through repeated comparisons of model simulations with observations in terms of a low-dimensional set of summary statistics. However, here we are interested in the joint inference of model parameters and the real rainfall $\boldsymbol{\xi}$, which makes ABC an inefficient approach.

To tackle this problem, we apply a Hamiltonian Monte Carlo (HMC) algorithm (Duane et al., 1987; Neal, 2011; Albert et al., 2016), which exploits the principles of Hamiltonian dynamics to attain very high acceptance rates and low auto-correlation even on high-dimensional sampling spaces. This makes it possible to explore such spaces with large steps, without compromising the acceptance rate, and thus making the algorithm very efficient. The inherent high efficiency of HMC is further boosted here by a time-scale separation analogous to the one described in Albert et al. (2016).

The HMC algorithm allows us to sample simultaneously from the posterior of Eq. 1 both model parameters $(\boldsymbol{\theta}, \sigma_\xi, \sigma_z)$ and realizations of the stochastic process $\boldsymbol{\xi}$. In particular, the rainfall potential $\boldsymbol{\xi}$ is inferred indirectly using prior knowledge, the observed runoff $\boldsymbol{Q}_{\mathrm{obs}}$, and the possibly inaccurate observed precipitation $\boldsymbol{P}_{\mathrm{obs}}$. The discharge data is used as an indirect source of knowledge about the rainfall, that complements the unreliable information due to the sparse rain-gauge measurements.

The method described here requires a stochastic input process, i.e., a rainfall model, a hydrological rainfall-runoff model to describe the observed discharges $\boldsymbol{Q}_{\mathrm{obs}}$, observation models for both rainfall and runoff, and prior probability distributions, which are all outlined below.

## 2.1 The rainfall model

The rainfall potential is described by a normal and linear Ornstein-Uhlenbeck (OU) process with mean zero and standard deviation unity, which can be written in the form of a Langevin equation as,

$$\dot{\xi}(t) = -\frac{\xi}{\tau} + \sqrt{\frac{2}{\tau}}\eta(t), \tag{2}$$

where $\eta(t)$ represents zero-mean Gaussian white noise and $\tau$ is the process correlation time. The latter is set equal to 636 seconds and will not be inferred. A list of model parameters that are assumed to be known and are not inferred is given in Table 1 at the end of Section 2.

The rainfall potential $\xi(t)$ is then transformed into real rain $P(t)$ by the nonlinear transformation (Sigrist et al., 2012; Ailliot et al., 2015; Del Giudice et al., 2016),

$$P(t) = r\big(\xi(t)\big) = \begin{cases} \lambda\big(\xi(t) - \xi_r\big)^{1+\gamma} & \text{if } \xi(t) > \xi_r \\ 0 & \text{if } \xi(t) \leq \xi_r \end{cases}, \tag{3}$$

where the zero/nonzero rain threshold $\xi_r$, the scaling factor $\lambda$ and the exponent $\gamma$ are all inferred parameters. A list of all inferred parameters is shown in Table 2 (see end of Section 2). The inherent rainfall stochasticity is thus accounted for by the stochastic process of Eq. 2, while the skewness of the rainfall distribution and a finite probability of zero rain are embedded in the model by the transformation of Eq. 3. Note that since $r(\xi) = 0$ for every $\xi \leq \xi_r$, the transformation $r$ is not invertible when the precipitation is zero. We will discuss this point in detail in Section 2.4.

## 2.2 The hydrological model

The stormwater runoff is modelled by a linear reservoir alimented by rainfall precipitations, $P(t)$, and a constant groundwater flow, $Q_{\text{gw}}$. The dynamics of the water volume $S(t)$ in the reservoir is thus governed by,

$$\dot{S}(t) = AP(t) + Q_{\text{gw}} - Q_M(t) \qquad \text{with} \qquad Q_M(t) = \frac{S(t)}{K}, \tag{4}$$

where $A$ is the estimated catchment area contributing to the rainfall-runoff dynamics, $Q_M$ is the hydrological model describing the runoff at the outlet of the system and $K$ is the retention time of the reservoir. It should be noted that the original hydrological model used in Del Giudice et al. (2016) is simplified here by omitting the daily variation due to the wastewater contribution. This is justified by the fact that in the present work we focus only on a single short dataset of 4 hours, whereas in Del Giudice et al. (2016) the authors consider 3 independent datasets covering a time span of about 48 days. One may refer to Section 4 for more details.

The discretized form of Eq. 4 reads as,

$$S_i = \left(1 - \frac{dt}{K}\right)S_{i-1} + \big(AP_{i-1} + Q_{\text{gw}}\big)dt, \tag{5}$$

where $S_i$ and $P_i$ are $S(t_i)$ and $P(t_i)$, respectively, with $i = 1, ..., N$. It should be noted that explicit methods, such as the forward Euler scheme applied in Eq. 5, are very easy to implement, however, they impose stringent limitations on the time step

size to ensure numerical stability. In general, explicit methods might not be sufficiently accurate in regions where the solution exhibits a rapidly varying behaviour. In that case it would be advisable to apply an implicit backward scheme, numerically more stable, albeit more difficult to implement. In the application discussed here, we reckon that the problem is simple enough to opt for the explicit forward scheme, thus trading off accuracy for an easier implementation. The intrinsic inaccuracy of the method is attenuated by choosing a discretization time step $dt$ that is sufficiently small compared to the system dynamics timescale.

The predicted discharge $Q_{M,i}(\boldsymbol{\xi},\boldsymbol{\theta}) = Q_M(t_i)$ $(i = 1, ..., N)$ can be calculated recursively using Eq. 5 with the initial condition $Q_{M,1}(\boldsymbol{\xi},\boldsymbol{\theta}) = S_1/K$. Straightforward calculations then yield,

$$Q_{M,i}(\boldsymbol{\xi},\boldsymbol{\theta}) = \frac{S_1}{K}\left(1 - \frac{dt}{K}\right)^{i-1} + A\frac{dt}{K}Q_{M,i}^{(\xi)} + \left[1 - \left(1 - \frac{dt}{K}\right)^{i-1}\right]Q_{\mathrm{gw}}, \tag{6}$$

where $Q_{M,i}^{(\xi)}$ is the $\xi$ - dependent contribution defined recursively as $Q_{M,1}^{(\xi)} = 0$ and

$$Q_{M,i}^{(\xi)} = (1 - \frac{dt}{K})Q_{M,i-1}^{(\xi)} + r(\xi_{i-1}) \qquad \text{with} \qquad i = 2, ..., N\,, \tag{7}$$

where we have used the rainfall potential transformation $P_i = r(\xi_i)$ of Eq. 3. The parameters of the hydrological model $K$, $Q_{\mathrm{gw}}$ and the initial water volume $S_1$ are unknown and to be inferred (see Table 2), while the catchment area is known and fixed as $A = 11815.8\ \mathrm{m}^2$ (Table 1).

## 2.3 Runoff observation model

The probability distribution for the observed discharges $\boldsymbol{Q}_{\mathrm{obs}}$ given the model predictions $Q_{M,i}(\boldsymbol{\xi},\boldsymbol{\theta})$ is assumed to be a normal error model with standard deviation $\sigma_z$, which reads as,

$$f(\boldsymbol{Q}_{\mathrm{obs}} \mid \boldsymbol{\xi},\boldsymbol{\theta},\sigma_z) \propto \exp\left[-(n_Q+1)\log(\sigma_z) - \sum_{s=1}^{n_Q+1}\frac{(H(Q_{\mathrm{obs},s}) - H(Q_{M,(s-1)j_Q+1}(\boldsymbol{\xi},\boldsymbol{\theta})))^2}{2\sigma_z^2}\right], \tag{8}$$

where $(s-1)j_Q+1$, with $s = 1, ..., n_Q+1$, are the indices corresponding to real observations in the discretized runoff dimension, and $H$ is a transformation introduced to take the heteroscedasticity of the errors into account (Del Giudice et al., 2016),

$$H(Q) = \beta\log\left(\sinh\left(\frac{\alpha+Q}{\beta}\right)\right), \tag{9}$$

with the parameters $\alpha = 25$ l/s and $\beta = 50$ l/s (Table 1). Since we are interested only in terms depending on the parameters to be inferred, in Eq. 8 we can neglect constant multiplicative factors such as the Jacobian $\prod_{s=1}^{n_Q+1}\frac{dH}{dQ}(Q_{\mathrm{obs},s})$.

## 2.4 Rainfall observation model

The observation error model for the rainfall, given the rainfall potential $\xi(t)$, is defined in the space of the rainfall potential as a normal distribution centered on $\xi(t)$ and with standard deviation $\sigma_\xi$. Its discretized form can be expressed in terms of the discretized potential $\boldsymbol{\xi}$ as a product of normal distributions,

$$f(\boldsymbol{\xi}_{\mathrm{obs}} \mid \boldsymbol{\xi},\sigma_\xi) = \prod_{s=1}^{n_P+1}\mathcal{N}_{(\xi_{(s-1)j_P+1},\sigma_\xi)}(\xi_{\mathrm{obs},s})\,, \tag{10}$$

where $\boldsymbol{\xi}_{\mathrm{obs}}$ is defined as the effective rainfall potential generating the observed rainfall and $\mathcal{N}_{(\xi,\sigma_\xi)}$ denotes a normal distribution with mean $\xi$ and standard deviation $\sigma_\xi$. Note that in Eq. 10 the $\xi_{(s-1)j_P+1}$ with $s = 1,...,n_P+1$ are the elements of the discretized potential $\boldsymbol{\xi}$ corresponding to time points where rainfall observations are available. This distribution is transformed to real rainfall by the inverse transformation $\xi_{\mathrm{obs}} = r^{-1}(P_{\mathrm{obs}})$ (Eq. 3). However, since all $\xi$-values below $\xi_r$ are transformed to zero rain, the transformation $r$ is invertible only where $P_{\mathrm{obs}} \neq 0$. Therefore, we need to distinguish two possible regimes, with and without rain:

– At time points where $P_{\mathrm{obs}} \neq 0$, the probability density of Eq. 10 reads,

$$f_{P\neq 0}\left(\boldsymbol{P}_{\mathrm{obs}} \mid \boldsymbol{\xi},\sigma_\xi\right) \propto \exp\left[-n_{P\neq 0}\log(\sigma_\xi) - \frac{\sum_{i,P\neq 0}\left(r^{-1}(P_{\mathrm{obs},i}) - \xi_{(i-1)j_P+1}\right)^2}{2\sigma_\xi^2}\right]\prod_{i,P\neq 0}\frac{1}{\mathcal{J}_i} \tag{11}$$

where the sum $\sum_{i,P\neq 0}$ extends only over time points $t_i$ where $P_{\mathrm{obs},i} \neq 0$ and $n_{P\neq 0}$ is the total number of such points. Moreover, the transformation from $\xi-$ to $P-$values requires the Jacobians $\mathcal{J}_i$ defined as,

$$\mathcal{J}_i = \frac{dr}{d\xi}\left(r^{-1}(P_{\mathrm{obs},i})\right) = \lambda(1+\gamma)\left(r^{-1}(P_{\mathrm{obs},i}) - \xi_r\right)^\gamma = \lambda(1+\gamma)\left(\frac{P_{\mathrm{obs},i}}{\lambda}\right)^{\frac{\gamma}{1+\gamma}}. \tag{12}$$

– At time points where $P_{\mathrm{obs}} = 0$ and therefore $\xi_{\mathrm{obs}}$ can take any value below $\xi_r$, we integrate the probability density of Eq. 10 over the interval $[-\infty, \xi_r]$. This yields the probability of zero observed rain, which turns out to be the cumulative distribution function of the normal distribution, that is,

$$f_{P=0}\left(\boldsymbol{P}_{\mathrm{obs}} \mid \boldsymbol{\xi},\sigma_\xi\right) = \prod_{i,P=0}\frac{1}{2}\left[1 + \mathrm{erf}\left(\frac{\xi_r - \xi_{(i-1)j_P+1}}{\sigma_\xi\sqrt{2}}\right)\right], \tag{13}$$

where the product $\prod_{i,P=0}$ extends over time points $t_i$ where $P_{\mathrm{obs},i} = 0$.

Therefore, $f\left(\boldsymbol{P}_{\mathrm{obs}} \mid \boldsymbol{\xi},\sigma_\xi\right) = f_{P\neq 0}\left(\boldsymbol{P}_{\mathrm{obs}} \mid \boldsymbol{\xi},\sigma_\xi\right) f_{P=0}\left(\boldsymbol{P}_{\mathrm{obs}} \mid \boldsymbol{\xi},\sigma_\xi\right)$.

## 2.5 The priors

At this point, the only elements of Eq. 1 that still need to be defined are the prior distributions $f(\boldsymbol{\theta},\sigma_\xi,\sigma_z)$ and $f(\boldsymbol{\xi})$. The former is simply the product of normal or log-normal univariate probability densities for each individual parameter to be inferred. The parameter vector $\boldsymbol{\theta}$ here includes the parameters of the hydrological model, $K$, $Q_{\mathrm{gw}}$ and $S_1$, as well as those of the transformation $r$ (Eq. 3), that is, $\xi_r$, $\lambda$ and $\gamma$. We infer a total of 8 parameters, listed in Table 2. For the parameters $S_1$ and $\xi_r$ the prior densities are assumed to be normal distributions, whereas for the other parameters ($K$, $Q_{\mathrm{gw}}$, $\lambda$ and $\gamma$) the prior densities are assumed to be log-normal distributions,

$$f(\theta) \propto \exp\left[-\log\theta - \frac{(\log\theta - \mu_{\mathrm{LN}})^2}{2\sigma_{\mathrm{LN}}^2}\right], \tag{14}$$

with mean and standard deviation $\mu_{\mathrm{LN}}$ and $\sigma_{\mathrm{LN}}$, respectively. Analogous log-normal distributions are assumed for the rainfall and discharge observation uncertainties, $\sigma_\xi$ and $\sigma_z$, respectively. The prior distributions for all parameters to be inferred, with their mean values and standard deviations, are summarized in Table 2.

Our prior knowledge of the rainfall potential $\boldsymbol{\xi}$ is defined in terms of a function $\mathcal{S}(\boldsymbol{\xi})$, called *action* (Lau and Lubensky, 2007; Albert et al., 2016), as,

$$f(\boldsymbol{\xi}) \propto \exp\left[-\mathcal{S}(\boldsymbol{\xi}) - \xi_1^2/2\right], \tag{15}$$

where the action can be written in its discretized form for the SDE model of Eq. 2 as (see Appendix A),

$$\mathcal{S}(\boldsymbol{\xi}) \approx \frac{\xi_N^2}{4} - \frac{\xi_1^2}{4} + \sum_{i=2}^{N}\left[\frac{\tau}{4dt}\left(\xi_i - \xi_{i-1}\right)^2 + \frac{dt}{4\tau}\xi_i^2\right], \tag{16}$$

and the initial condition for $\boldsymbol{\xi}$ is specified as the marginal distribution of a standard OU process, which is a standard normal distribution for $\xi_1$. It is noteworthy that the HMC method described here always requires an explicit analytical form for the action $\mathcal{S}(\boldsymbol{\xi})$, which is essentially just the negative log of the prior density $f(\boldsymbol{\xi})$, which is also needed in any other Metropolis-type sampling algorithm. Although in this study we follow the approach of Del Giudice et al. (2016) and resort to a linear OU process as a precipitation generator, an analytical expression for the action is generally readily available even for more complex and nonlinear SDE models. More details about the procedure to derive the action for generic SDE models can be found in Appendix A.

**Table 1.** List of model parameters that are assumed to be known, with their values and units.

| Parameter | Value | Units | Description |
|:---:|:---:|:---:|:---:|
| $\tau$ | 636 | $s$ | Autocorrelation time of the stochastic process $\xi(t)$ |
| $A$ | 11815.8 | $m^2$ | Catchment area |
| $\alpha$ | 25 | $l/s$ | Coefficient of transformation $H(Q)$ |
| $\beta$ | 50 | $l/s$ | Coefficient of transformation $H(Q)$ |

Before setting off to implement the HMC algorithm, we take one further convenient step, i.e., we apply the transformation from the coordinates $\xi$ to the so-called staging variables $u$ (Tuckerman et al., 1993) using,

$$u_{sj_P+1} = \xi_{sj_P+1} \qquad \text{with} \qquad s = 0,...,n_P, \tag{17}$$

which leaves the components corresponding to measurement points unchanged, and

$$u_{sj_P+k} = \xi_{sj_P+k} - \xi_{sj_P+k}^* \qquad \text{with} \qquad s = 0,...,n_P-1 \quad \text{and} \quad k = 2,...,j_P, \tag{18}$$

and with,

$$\xi_{sj_P+k}^* = \frac{(k-1)\xi_{sj_P+k+1} + \xi_{sj_P+1}}{k}, \tag{19}$$

for the intermediate discretization points. Also relevant are the inverse transformations for the discretization points,

$$\xi_{sj_P+k} = \sum_{l=k}^{j_P+1} \frac{k-1}{l-1}u_{sj_P+l} + \frac{j_P-k+1}{j_P}u_{sj_P+1} \quad \text{with} \quad s = 0,...,n_P-1 \quad \text{and} \quad k = 2,...,j_P. \tag{20}$$

**Table 2.** Prior probability densities for the inferred parameters, with their mean values ($\mu$), standard deviations ($\sigma$) and units. N and LN stand for normal and log-normal distributions, respectively. Note that here, also for log-normal distributions, $\mu$ and $\sigma$ refer to the mean value and standard deviation of a parameter, not its log. The mean and standard deviation for the log, $\mu_{\mathrm{LN}}$ and $\sigma_{\mathrm{LN}}$, respectively, are given by $\mu_{\mathrm{LN}} = \log(\mu^2/\sqrt{\mu^2 + \sigma^2})$ and $\sigma_{\mathrm{LN}} = \sqrt{\log(1 + \sigma^2/\mu^2)}$.

| Parameter | Prior distribution | $\mu$ | $\sigma$ | Units | Description |
|---|---|---|---|---|---|
| $K$ | LN | 284.4 | 57.6 | $s$ | Retention time |
| $Q_{\mathrm{gw}}$ | LN | 6 | 1 | $l/s$ | Groundwater flow |
| $\sigma_z$ | LN | 4.5 | 0.45 | $l/s$ | Runoff observation uncertainty |
| $\sigma_\xi$ | LN | 0.65 | 0.3 | - | Rainfall observation uncertainty |
| $\lambda$ | LN | 0.1/60 | 0.05/60 | $l/(s \cdot m^2)$ | Scaling factor of transformation $r(\xi)$ |
| $\gamma$ | LN | 0.5 | 0.25 | - | Exponent of transformation $r(\xi)$ |
| $\xi_r$ | N | 0.5 | 0.1 | - | Zero/nonzero rain threshold |
| $S_1$ | N truncated to interval $[0, \infty)$ | 0 | 5000 | $l$ | Initial water volume |

The action $\mathcal{S}(\boldsymbol{\xi})$ (Eq. 16) can be formulated in the space of $u$-coordinates (see Appendix B) as,

$$\mathcal{S}(\boldsymbol{u}) = \frac{k\tau}{4(k-1)dt} \sum_{s=1}^{n_P} \sum_{k=2}^{j_P} u_{(s-1)j_P+k}^2 + \frac{\tau}{4j_P dt} \sum_{s=1}^{n_P} (u_{(s-1)j_P+1} - u_{sj_P+1})^2$$

$$+ \frac{u_N^2}{4} - \frac{u_1^2}{4} + \frac{dt}{4\tau} \sum_{s=1}^{n_P} \left[ u_{sj_P+1}^2 + \sum_{k=2}^{j_P} \left( \sum_{l=k}^{j_P+1} \frac{k-1}{l-1} u_{(s-1)j_P+l} + \frac{j_P-k+1}{j_P} u_{(s-1)j_P+1} \right)^2 \right], \tag{21}$$

while the initial condition $\xi_1^2/2$ can be simply replaced by $u_1^2/2$. This coordinate transformation, analogous to a transformation to canonical coordinates, is not a strict requirement of the HMC method and undoubtedly adds a further degree of complexity to the overall strategy. However, it also bears remarkable computational benefits. Indeed, the first term on the right-hand side

of Eq. 21 describes the potential of a system of uncoupled harmonic oscillators, which can be effortlessly solved analytically. These dynamics also turn out to be much faster than all other characteristic timescales of the system. In Section 3 we will describe in detail how this can be exploited.

## 3 The HMC algorithm

The HMC algorithm interprets the negative logarithm of the posterior density as a potential energy driving the dynamics

of a fictitious statistical mechanics system whose configurations, namely, the system's degrees of freedom, are described by combinations of both parameters $(\boldsymbol{\theta}, \sigma_\xi, \sigma_z)$ and realizations $\boldsymbol{\xi}$ of the stochastic process $\xi(t)$. The degrees of freedom of the system are coupled to conjugate variables, that is, to *momenta* $\boldsymbol{\pi}$, paired with the parameters $(\boldsymbol{\theta}, \sigma_\xi, \sigma_z)$, and $\boldsymbol{p}$, paired with the realizations $\boldsymbol{\xi}$. The posterior density of Eq. 1 can be rewritten in the following discretized form,

$$f(\boldsymbol{\xi}, \boldsymbol{\theta}, \sigma_\xi, \sigma_z \mid \boldsymbol{P}_{\mathrm{obs}}, \boldsymbol{Q}_{\mathrm{obs}}) \propto \int \exp\left[-\mathcal{H}_{\mathrm{HMC}}(\boldsymbol{\xi}, \boldsymbol{\theta}, \sigma_\xi, \sigma_z; \boldsymbol{\pi}, \boldsymbol{p})\right] d\boldsymbol{\pi} \, d\boldsymbol{p}, \tag{22}$$

with the Hamiltonian $\mathcal{H}_{\mathrm{HMC}}$ defined as,

$$\mathcal{H}_{\mathrm{HMC}}\left(\boldsymbol{\xi},\boldsymbol{\theta},\sigma_\xi,\sigma_z;\boldsymbol{\pi},\boldsymbol{p}\right) = -\log\left[f\left(\boldsymbol{\xi},\boldsymbol{\theta},\sigma_\xi,\sigma_z \mid \boldsymbol{P}_{\mathrm{obs}},\boldsymbol{Q}_{\mathrm{obs}}\right)\right] + \sum_{\alpha=1}^{N_p}\frac{\pi_\alpha^2}{2m_\alpha} + \sum_{i=1}^{N}\frac{p_i^2}{2m_i}, \tag{23}$$

where $N_p$ is the number of parameters to be inferred (8 in our case) and we have introduced two auxiliary terms on the right-hand side using the vectors of momenta $\boldsymbol{\pi}$ and $\boldsymbol{p}$, akin to the kinetic energies associated with the degrees of freedom of the fictitious statistical mechanics system.

The masses $m_\alpha$ and $m_i$ are tuning parameters of the algorithm. Since we want the coordinates $\xi_i$ corresponding to actual observations to be more tightly constrained than those corresponding to the intermediate discretization points, we set $m_{sj_P+1} = M$ with $s = 0,...,n_P$, for the "heavy" measurement points, and $m_{sj_P+k} = m$ with $s = 0,...,n_P-1$ and $k = 2,...,j_P$, for the "lighter" intermediate discretization points, with $M \gg m$.

The potential energy, proportional to $-\log\left[f\left(\boldsymbol{\xi},\boldsymbol{\theta},\sigma_\xi,\sigma_z \mid \boldsymbol{P}_{\mathrm{obs}},\boldsymbol{Q}_{\mathrm{obs}}\right)\right]$, guarantees that each state of the system is compatible with the observations, and constrained within their measurement uncertainties, as well as with the prior distributions for both model parameters and rainfall potential realizations $\boldsymbol{\xi}$.

The HMC algorithm iterates the following steps. First, vectors of momenta $\boldsymbol{\pi}$ and $\boldsymbol{p}$ are drawn from the zero-mean normal distributions defined by the kinetic terms in Eq. 23. Then, the system is let to evolve for a fictitious time interval $d\tau$ in the $(\boldsymbol{\xi},\boldsymbol{\theta},\sigma_\xi,\sigma_z;\boldsymbol{\pi},\boldsymbol{p})$ phase space according to Hamilton's equations. This Hamiltonian dynamics is controlled by tuning the fictitious masses in Eq. 23, which represent the variances of the normal distributions for the corresponding momenta. Finally, the discretization error on the energy conservation introduced with the integration of Hamilton's equations is corrected by a Metropolis accept/reject step. The resulting marginal distributions of the Markov chains of the system configurations $(\boldsymbol{\xi},\boldsymbol{\theta},\sigma_\xi,\sigma_z)$ represent a sample of the sought posterior density. The method is described in detail in Albert et al. (2016).

Note that the presence of pronounced local minima in a high-dimensional phase space might represent an insurmountable obstacle even for more refined implementations of the HMC method, e.g., with automatic tuning of the algorithm hyper-parameters (see Section 4.1 for more details). In that case one would have to resort to further enhancements such as Metadynamics (Laio and Gervasio, 2008) or Parallel Tempering (Swendsen and Wang, 1986; Marinari and Parisi, 1992; Earl and Deem, 2005). However, this is not an issue in the specific application discussed here.

Using Eq. 1 and the definitions of Sections 2.1, 2.2, 2.3, 2.4 and 2.5, we write the HMC Hamiltonian as,

$$\mathcal{H}_{\mathrm{HMC}} = \mathcal{H}_N + \mathcal{H}_n + \mathcal{H}_1, \tag{24}$$

where the three components describe dynamics occurring on different timescales. Let us consider them individually. The first component,

$$\mathcal{H}_N = \sum_{s=1}^{n_P}\sum_{k=2}^{j_P}\left[\frac{p_{(s-1)j_P+k}^2}{2m} + \frac{k\tau}{4(k-1)dt}u_{(s-1)j_P+k}^2\right], \tag{25}$$

contains the harmonic term for the intermediate discretization points from the action of Eq.21 and scales like the total number of discretization points $N$. The second component,

$$
\begin{aligned}
\mathcal{H}_n = {} & \sum_{s=1}^{n_P+1} \frac{p_{(s-1)j_P+1}^2}{2M} + \sum_{s=1}^{n_P} \frac{\tau}{4 j_P dt} \left( u_{(s-1)j_P+1} - u_{sj_P+1} \right)^2 \\
& + \sum_{s=1}^{n_Q+1} \frac{(H(Q_{\text{obs},s}) - H(Q_{M,(s-1)j_Q+1}(\boldsymbol{u},\boldsymbol{\theta})))^2}{2\sigma_z^2} + (n_Q+1)\log(\sigma_z) \\
& + \sum_{i,P\neq 0} \left[ \frac{(r^{-1}(P_{\text{obs},i}) - u_{(i-1)j_P+1})^2}{2\sigma_\xi^2} + \log \mathcal{J}_i \right] + n_{P\neq 0}\log(\sigma_\xi) - \sum_{i,P=0} \log\left[ \frac{1}{2}\left( 1 + \text{erf}\left( \frac{\xi_r - u_{(i-1)j_P+1}}{\sigma_\xi \sqrt{2}} \right) \right) \right],
\end{aligned} \tag{26}
$$

contains a harmonic term for the measurement points from Eq.21 and the observation error models $f(\boldsymbol{Q}_{\text{obs}} \mid \boldsymbol{u},\boldsymbol{\theta},\sigma_z)$ and $f(\boldsymbol{P}_{\text{obs}} \mid \boldsymbol{u},\sigma_\xi)$ from Sections 2.3 and 2.4, respectively. All the components of Eq. 26 scale like the number of observations $n_P$ or $n_Q$. Note that both the observation models for runoff and rainfall are rewritten in the space of $u$-coordinates. While the coordinate transformation $\boldsymbol{\xi} \to \boldsymbol{u}$ is straightforward for the rainfall observation model, which depends only on measurement points, it is somewhat more cumbersome for the runoff observation model, which requires the $\xi$-dependent component $Q_{M,i}^{(\xi)}$ (see Eqs. 6 and 7) to be rewritten in the $u$-space as $Q_{M,i}^{(u)}$. Such transformation is described in Appendix C. The third component of the Hamiltonian $\mathcal{H}_{\text{HMC}}$ is,

$$
\begin{aligned}
\mathcal{H}_1 = {} & \sum_{\alpha=1}^{N_p} \frac{\pi_\alpha^2}{2m_\alpha} + \frac{u_N^2}{4} + \frac{u_1^2}{4} + \frac{dt}{4\tau} \sum_{s=1}^{n_P} \left[ u_{sj_P+1}^2 + \sum_{k=2}^{j_P} \left( \sum_{l=k}^{j_P+1} \frac{k-1}{l-1} u_{(s-1)j_P+l} + \frac{j_P-k+1}{j_P} u_{(s-1)j_P+1} \right)^2 \right] \\
& + \sum_{\theta \in \boldsymbol{\theta}_\text{N}} \frac{(\theta - \mu_\theta)^2}{2\sigma_\theta^2} + \sum_{\theta \in \boldsymbol{\theta}_\text{LN}} \left[ \log\theta + \frac{(\log\theta - \mu_{\text{LN},\theta})^2}{2\sigma_{\text{LN},\theta}^2} \right],
\end{aligned} \tag{27}
$$

which includes the remaining terms from the rainfall potential prior (Eq. 15) and the model parameter priors. The latter, i.e., the last two terms in Eq. 27, are sums over the parameters $\boldsymbol{\theta}_\text{N} = (S_1, \xi_r)$ and $\boldsymbol{\theta}_\text{LN} = (K, Q_{\text{gw}}, \lambda, \gamma, \sigma_\xi, \sigma_z)$, whose prior densities are normal and log-normal distributions, respectively. Note the sign change in front of the term $u_1^2/4$ with respect to Eq. 21 due to the initial condition for $u_1$ (see Eq. 15). The component $\mathcal{H}_1$ does not grow unbounded with neither $N$ nor $n_P$ or $n_Q$ (notice that $dt \propto 1/N$).

Let us now exploit the different timescales characterizing the three components of the Hamiltonian $\mathcal{H}_{\text{HMC}}$. We assume the regime $\mathcal{H}_N \gg \mathcal{H}_n \approx \mathcal{H}_1$, implying that the number of data points is not too large and that the total number of discretization points is instead large compared to the number of data points. Under these conditions the dynamics of the system occur on very different time scales. In particular, the dynamics described by $\mathcal{H}_N$ are much faster than the other components of the Hamiltonian and therefore impose the most stringent limitations on the step size for the numerical integration of Hamilton equations. However, we use Trotter's formula (Tuckerman et al., 1992) to construct a reversible molecular dynamics integrator to take the different time scales into account as described in Albert et al. (2016). In particular, we exploit the fact that the much faster dynamics of the intermediate discretization points described by $\mathcal{H}_N$ is analogous to a system of uncoupled harmonic oscillators that can be solved analytically. This analytical solution gives a significant boosting contribution to the intrinsic

efficiency of the HMC algorithm. The "slow" remaining components of the Hamiltonian, $\mathcal{H}_n$ and $\mathcal{H}_1$, can be integrated using a much larger integration step, which allows us to explore the high-dimensional parameter space of the system with remarkable efficiency.

As explained in Albert et al. (2016), the decoupling of the different dynamics and the analytical solution of the expensive fast component is always possible for one-dimensional SDE models. It is also possible in the case of multiple independent variables, where the decoupling procedure can be applied to each of them individually. In this way, our HMC approach covers a significant range of possible hydrological modelling scenarios. However, in this work we focus only on 1D models, leaving the exploration of higher-dimensional models to future studies.

## 4    Case study

In this work we apply a HMC method with a stochastic input model (SIP) following Del Giudice et al. (2016) to investigate the rainfall-runoff dynamics of an urban catchment based on real rainfall and runoff observations. The catchment is a small and fast-reacting sewer network in Adliswil, near Zurich, Switzerland. Two typical scenarios of possible rainfall data are considered here, that is, we use: 1) high-resolution data, with a time resolution of 1 minute, recorded by two pluviometers (denoted $P1_a$

and $P1_b$) located in the immediate vicinity of the catchment area, 2) low-resolution data, with a time resolution of 10 minutes, recorded by a pluviometer (denoted P2) located further away from the catchment, at a distance of about 6 Km. We shall refer to the two scenarios as scenario 1 (Sc1) and scenario 2 (Sc2), respectively. More information can be found in Del Giudice et al. (2016).

The precipitation data $\boldsymbol{P}_{\text{obs}}$ used in this study contain $n_P + 1 = 241$ observations in Sc1 and $n_P + 1 = 25$ observations in

Sc2, covering a total observation time $T = 240$ minutes. The initial and final observation time points are the same for both scenarios and include a storm event that took place in the evening of June 10, 2013, approximately between 18:00 and 20:00. We should mention here a substantial difference with respect to Del Giudice et al. (2016). In their work, the authors base the inference process on three independent time series covering a total observation time of 1710 minutes distributed over a time span of 48 days. In the work presented here, instead, we consider only the first of the 3 time-series, which happens to be also the

shortest. Although the HMC algorithm could deal with the multiple independent time-series used by Del Giudice et al. (2016), our simplification finds its justification in that it reduces the computational requirements of the problem, without compromising the goal of our work, that is, showing the feasibility of Bayesian inference of both model parameters and high-dimensional system states (i.e., the $\boldsymbol{\xi}$'s) under the considerable hardship of a stochastic input process.

The discharge flow at the outlet of the catchment was measured with a time resolution of 4 minutes, and the output obser-

vations $\boldsymbol{Q}_{\text{obs}}$, containing $n_Q + 1 = 61$ measurements, are the same for both scenarios and considered accurate compared to the precipitation data. The initial and final observation times are the same as for the input time-series. The time-series for the observed outputs ($\boldsymbol{Q}_{\text{obs}}$) as well as the observed precipitation ($\boldsymbol{P}_{\text{obs}}$) for both Sc1 and Sc2 are shown in Fig. 5.

Scenario 1 represents a best-case scenario of input data availability, and we shall therefore classify it as the accurate input scenario, while scenario 2 is a typical example of inaccurate and unreliable input data due to both its sparsity and the distance

of the rain-gauge P2 from the area of interest. The runoff observations $Q_{\text{obs}}$ exhibit an important rainfall event (the storm) represented by an evident discharge peak. While this event is properly recorded by the rain-gauges of Sc1, the inaccurate input data of Sc2 misleadingly recorded the event at an earlier time period, presumably when the storm passed over the location of the distant pluviometer P2.

We are particularly interested in the performance of the combined SIP-HMC method in the latter case, characterized by faulty precipitation data, which clearly represents the most challenging scenario and therefore the hardest test for the HMC method. Therefore, our work consists of three main steps. First, we use the combined SIP-HMC approach described above, with the inaccurate precipitation data $P_{\text{obs}}$ of Sc2, to calibrate the model and infer the unknown "true" average rainfall pattern over the catchment. Then, we use the accurate rainfall observations from Sc1 as a reference to assess the quality of the simulated "true" rain. Finally, we repeat the calibration process using the accurate data of Sc1 as a further validation for the method.

## 4.1 Implementation

The HMC algorithm is implemented in C++14 using the open-source ADEPT library (Automatic Differentiation using Expression Templates, version 1.1) for the calculation of the gradients of the Hamiltonian $\mathcal{H}_{\text{HMC}}$ (Hogan, 2014). Automated differentiation allows us to automatize the algorithm to a large extent, thus making it applicable to a broad range of models with relatively little implementation effort. Indeed, for the hydrological application presented here we resorted to the algorithm already implemented for the simpler toy model studied in Albert et al. (2016). The implementation of the algorithm remained essentially unaltered, except only for the Hamiltonian $\mathcal{H}_{\text{HMC}}$ that had to be modified according to Eqs. 25, 26 and 27.

The simulations were run on 2.6-3.7 GHz processors Xeon-Gold 6142 with 196 GB of memory. We observed a relatively short burn-in phase for all inferred parameters, suggesting the possibility of a straightforward (embarrassingly simple) parallelization of the algorithm obtained by simply breaking up the Markov chains into smaller independent chains that can then be executed as parallel processes. It is well-known that Markov Chain Monte Carlo (MCMC) methods, like the HMC algorithm employed here, are indeed very well suited for parallel computing. This kind of approach was proven successfully in Albert et al. (2016) with a toy system. In that case we used an OpenMP-based parallel implementation of the algorithm and observed a reasonably linear strong scaling behaviour with up to 16 parallel processes.

In the present work, for Sc2, after an initial single burn-in chain of 75k steps, which is then disregarded, we limited ourselves to running 4 independent Markov chains each of length 100k steps based on a serial implementation of the algorithm. For Sc1, which is faster than Sc2 due to the much smaller number of intermediate discretization points (see below for more details), we considered a single chain of 750k steps and disregarded the first 150k steps. The extension of the current serial version to an OpenMP parallel implementation of the HMC code would be straightforward.

We set a fine-grid time step $dt = 10$ seconds and a total number of discretization points $N = 1441$ for both scenarios. It is easy to verify that these conditions are fulfilled by setting the number of bins between consecutive observations to $j_P = 6$ or $j_P = 60$ on the precipitation dimension for Sc1 and Sc2, respectively, and $j_Q = 24$ for the runoff dimension. The initial configuration of the system state $\boldsymbol{\xi}$ for the burn-in chain is a random realization of the OU process of Eq. 2, while the parameters are set equal to the mean values of their prior distributions (see Table 2).

The algorithm requires tuning of two sets of parameters, that is, the parameters defining the Hamiltonian propagator in the molecular dynamics part of the HMC algorithm (see Eq. 26 in Albert et al. (2016)), and the masses $m$, $M$ and $m_\alpha$ defining the kinetic energy of the system in $\mathcal{H}_{\text{HMC}}$. Thus, in the Hamiltonian propagator we set the integration time $\Delta\tau = 0.015$ and the number of integration steps $P = 3$, while the masses were set to $m = 0.4$ for the intermediate discretization points and $M = 1.6$ for the measurement points. The masses for the inferred parameters are given in Table 3.

It should be noted that the integration time interval of the Hamiltonian propagator could be automatically optimized by employing the so-called No-U-Turn Sampler (NUTS) (Hoffman and Gelman, 2014) and the masses of the kinetic terms could be tuned by adapting their values to the curvature of the energy landscape (Girolami and Calderhead, 2011; Hartmann et al., 2022). The efficiency of the HMC algorithm would surely benefit from these approaches, but at some cost in terms of implementation efforts. In this work we simply opt for a manual tuning of the hyper-parameters mentioned above.

In Sc2, a full Markov chain of 100k steps requires approximately 1 hour and 20 minutes on our hardware, while a chain of 750k in Sc1 requires about 2.5 hours. At each iteration of the chain the algorithm infers 1449 parameters, that is, 8 model parameters $(\boldsymbol{\theta}, \sigma_\xi, \sigma_z)$ plus $N = 1441$ coordinates of the system state $\boldsymbol{\xi}$.

The algorithm spends $\approx 96\%$ of the total run time in the molecular dynamics part, that is, in the loop for the integration of Hamilton's equations. The loop is called once in each step of the Markov chain. At each call, the HMC algorithm performs one evaluation of the posterior function $\mathcal{H}_{\text{HMC}}$, for the calculation of the system energy, and 6 evaluations of its derivatives (with $P = 3$ integration steps). In our case study and implementation of the algorithm, each differentiation of the posterior function turns out to be about 6 times more expensive than its plain evaluation. Although this factor is only indicative and may vary to some extent between different implementations, the calculation of the derivatives represents in general the major bottleneck in the performance of the HMC algorithm.

**Table 3.** Parameter masses $m_\alpha$ for the kinetic term of Eq. 27.

| Parameter | $K$ | $Q_{\text{gw}}$ | $\sigma_z$ | $\sigma_\xi$ | $\lambda$ | $\gamma$ | $\xi_r$ | $S_1$ |
|---|---|---|---|---|---|---|---|---|
| **Mass** | $10^{-5}$ | $1.0$ | $1.0$ | $1.0$ | $2 \cdot 10^5$ | $0.5$ | $15.0$ | $10^{-7}$ |

## 4.2 Results

The Markov chains for the model parameters generated using the unreliable input data of Sc2 are shown in Fig. 1. A simple visual inspection leads us to conclude with a good confidence that the chains have appropriately converged and the mixing in parameters space is satisfactory. Figure 2 shows the Markov chains for the same model parameters, generated using the accurate rainfall observations of Sc1. The corresponding marginal posterior probability densities for the model parameters, for both Sc2 and Sc1, are shown in Fig. 3 together with the initial prior distributions.

The two scenarios bear some interesting albeit not surprising differences. In general, the posterior distributions generated in Sc1 tend to be narrower than the corresponding distributions in Sc2, clearly reflecting the accuracy of the precipitation data. The rainfall observational error $\sigma_\xi$ appears to be also strongly shifted to lower values in Sc1, indicative of the reduced uncertainty

compared to the faulty Sc2. On the other hand, the practically identical posterior densities for the runoff observational error $\sigma_z$ reflect the fact that the discharge data $\boldsymbol{Q}_{\text{obs}}$ are the same for both scenarios.

Moreover, Figure 3 shows that among the rain-related parameters of the transformation of Eq. 3, the marginal posterior for the exponent $\gamma$ exhibits the largest shift towards smaller values when going from Sc2 to Sc1, while the zero/nonzero rain threshold $\xi_r$ is only mildly shifted to larger values and the scaling factor $\lambda$ seems to be essentially unaltered, besides the narrowing effect due to the improved accuracy of the precipitation data. The algorithm automatically tunes the parameters of the rainfall potential transformation to match the available rainfall data. The smaller value of the exponent $\gamma$ in Sc1 compared

to Sc2 reflects exactly this attempt of the algorithm to find a better fit to the rain observations, especially where precipitation values are large.

Among the parameters of the hydrological model, the marginal distribution of the retention time $K$ appears to be clearly shifted to smaller values in Sc1, suggesting a faster reacting system compared to Sc2, while the groundwater contributions $Q_{\text{gw}}$ and the initial water volume $S_1$ exhibit only very minor shifts.

In Fig. 4 we also show two typical Markov chains from Sc2 for the stochastic process $\boldsymbol{\xi}$, evaluated at two time-points with and without rain. The two chains clearly fluctuate above (point with rain) and below (point without rain) the inferred zero/nonzero rain threshold $\xi_r$. Analogously to the model parameters, the Markov chains for $\boldsymbol{\xi}$ appear to have converged and to be well mixed. The $\boldsymbol{\xi}$ chains in Sc1 are qualitatively identical to those of Sc2 and are not shown here.

In the left panels of Fig. 5 we compare the inferred discharge and rainfall patterns, $\boldsymbol{Q}_M(\boldsymbol{\xi}, \boldsymbol{\theta})$ and $r(\boldsymbol{\xi})$, respectively, based

on the inaccurate rainfall data of Sc2, with the observed runoff $\boldsymbol{Q}_{\text{obs}}$ and precipitation $\boldsymbol{P}_{\text{obs}}$. The measured outflow (upper panel, open red squares) clearly exhibits discharge peaks that are coupled to corresponding peaks in the observed rainfall in Sc1 (lower panel, open purple squares) but not in Sc2 (lower panel, filled blue squares). These "missing" peaks are indications that some rainfall events were either not detected, or recorded at a different time point, by the rain-gauge (P2) that produced those data. This is of course in line with the inaccuracy of Sc2.

The observed output peaks are used by the HMC algorithm as an additional source of information about the rain falling over the catchment area during the observation time. This new information, together with the stochastic input model, is used to attempt a reconstruction of a true rainfall pattern. The simulated rainfall and outflow patterns are represented by the medians of their inferred distributions (black line) and an uncertainty given by the 2.5%-97.5% quantiles (gray area). The rainfall pattern reconstructed using the inaccurate data of Sc2 (lower left panel) clearly displays the peaks corresponding to the rainfall events

that had been missed by the pluviometer P2 located away from the catchment (filled blue squares). Such predicted peaks reproduce very accurately in both time and duration the rainfall events detected in Sc1 by the rain-gauges P1$_{\mathbf{a}}$ and P1$_{\mathbf{b}}$ in the proximity of the area of interest (open purple squares).

The right panels of Figure 5 compare runoff and precipitation observations with the inferred discharge and rainfall patterns in the accurate framework of Sc1. Although the simulated rain in Sc1 features somewhat less intense peaks than in Sc2, the

465 remarkable fact that emerges from the comparison of the left and right panels of Figure 5 (referring to Sc2 and Sc1, respectively) is that the rainfall patterns predicted in the two scenarios are qualitatively very similar, despite the significant difference in the accuracy of the data used for the inference.

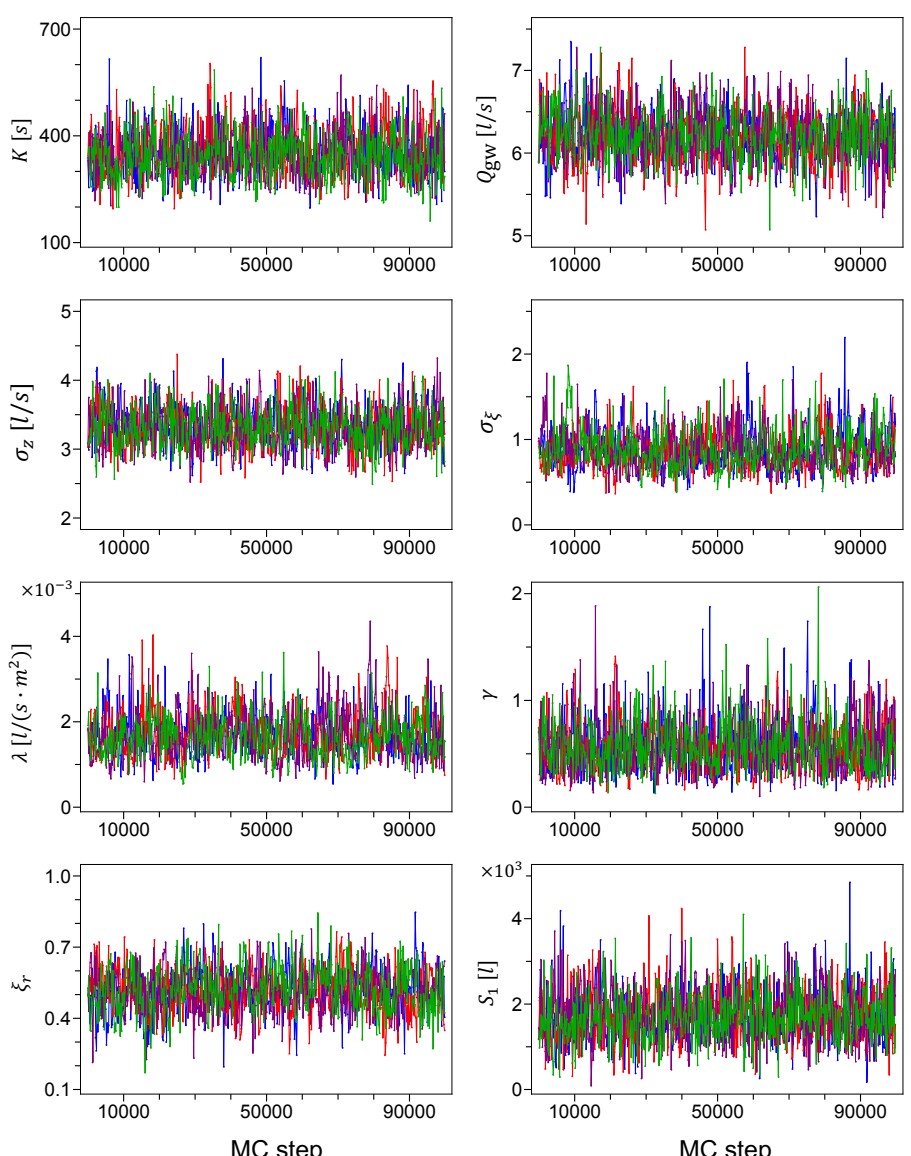

**Figure 1.** Markov chains for the model parameters generated using the faulty input data of Sc2. As explained in Section 4.1, we have 4 independent chains for each parameter.

Although not shown here, we have also run the HMC inference without rainfall data at all, i.e., omitting the term $f\left(\boldsymbol{P}_{\mathrm{obs}} \mid \boldsymbol{\xi}, \sigma_{\xi}\right)$ in the posterior density, obtaining both model parameter marginals and a predicted rainfall pattern that are substantially identical to those obtained with the inaccurate data of Sc2. Essentially, in Sc2 the HMC algorithm "learns" that the observed rain is unreliable and should thus be ignored. However, in most applications the accuracy and reliability of the measured precipitation

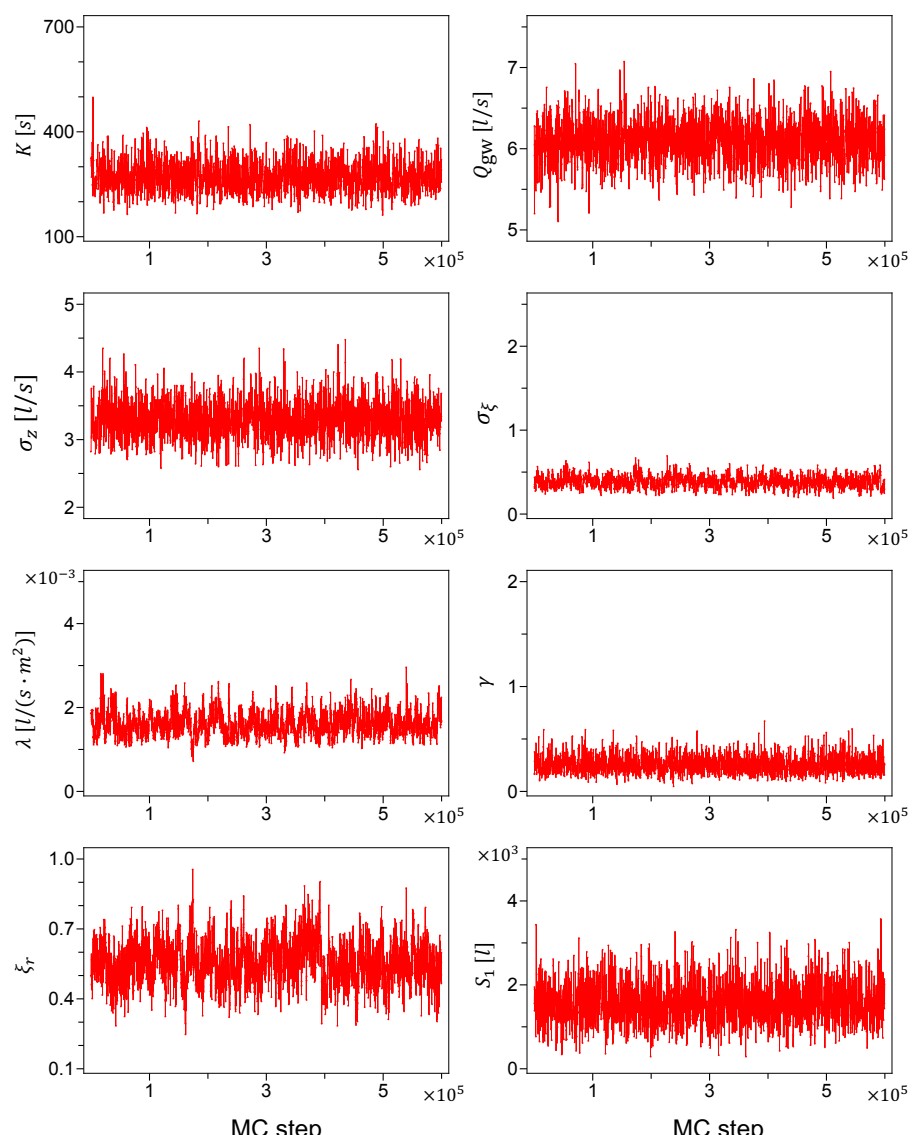

**Figure 2.** Markov chains for the model parameters from Sc1. Unlike Sc2, we have in this case a single chain for each parameter.

data is unknown a priori. In those cases the rainfall observations can be safely used in the inference process, since the algorithm itself will assess its accuracy and possibly disregard it in favor of a more reliable reconstructed rainfall.

The inferred outflows, shown in the upper panels of Figures 5 match very well the observations and are essentially identical, 475 compatibly with the assumption that runoff data are accurate and the same for the two scenarios.

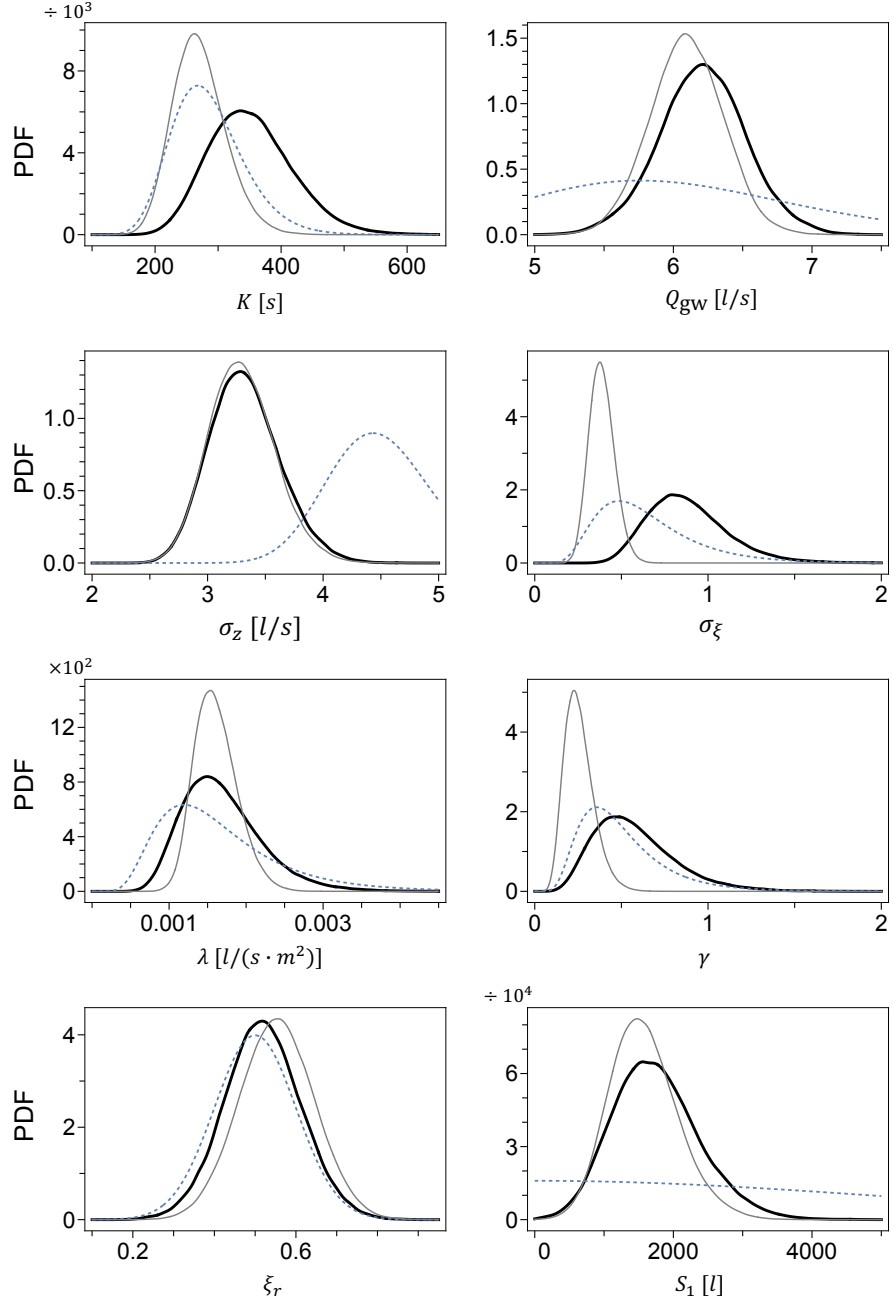

**Figure 3.** Marginal posterior probability densities for the model parameters from Sc2 (thick black line) and Sc1 (thin gray line). The dashed lines represent the prior densities.

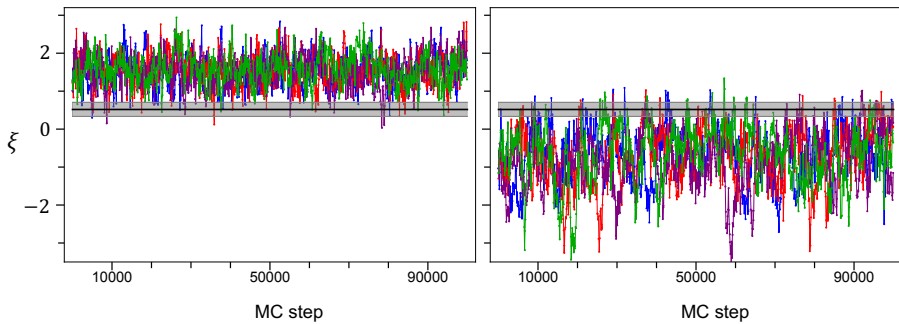

**Figure 4.** Typical Markov chains for two different points of the stochastic process $\boldsymbol{\xi}$ from Sc2, with (left) and without (right) rain. The horizontal solid line is the median of the inferred zero/nonzero rain threshold $\xi_r$, while the shaded area represents its uncertainty given by the 2.5 and 97.5 percentiles. As in Fig. 1, we have 4 independent chains for each point.

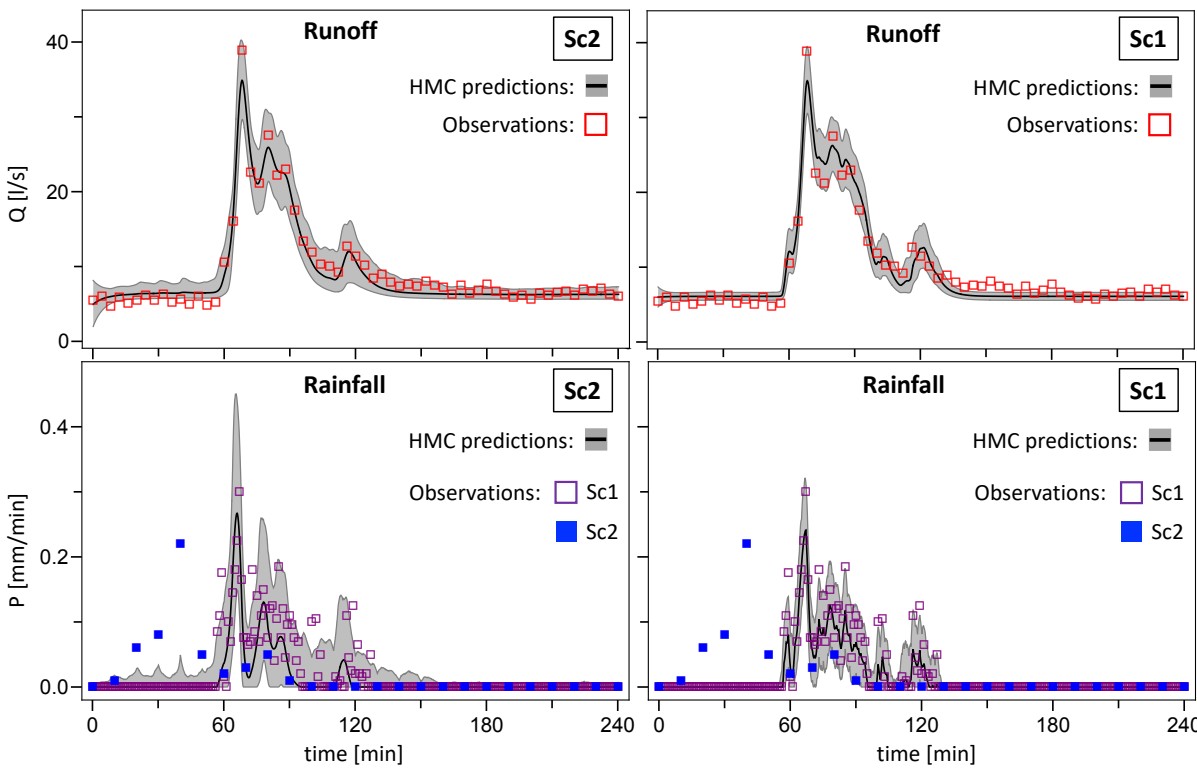

**Figure 5.** Comparison of observed and predicted discharges (top) and rainfall (bottom), based on Sc2 (left) and Sc1 (right) data. Predictions are represented by the median (black line) of the corresponding inferred distributions and a confidence interval given by the 2.5 and 97.5 percentiles (gray area). In the bottom panels the predicted rain is compared with both the inaccurate data of Sc2 (filled squares) and the accurate data of Sc1 (open squares).

## 5    Conclusions

The goal of this work is to demonstrate that HMC algorithms employing a time-scale separation can solve hard inference problems with stochastic hydrological models. In Albert et al. (2016) we proposed a novel implementation of an HMC algorithm combined with a multiple time-scale integration for Bayesian parameter inference with nonlinear SDE models, and we demonstrated the performance of the method using a rainfall-runoff toy model with synthetic data time-series. This work is the first application to a real-world case study, with real time-series of measured rainfall and outflows. We show that the HMC algorithm is a powerful inference method for the data-driven calibration of hydrological model parameters, especially well-suited for both computationally expensive stochastic models and cases, far from rare in hydrology, where the precipitation data is inaccurate and unreliable.

The combined SIP-HMC method presented here allows us to estimate probabilistically and with great accuracy the "true" average rain input to a hydrological system also in case of highly inaccurate precipitation data, using only prior knowledge and the observed outflow. Runoff data are used by the algorithm as a first-hand information resource about the unknown precipitation over the catchment. This information can override the available, and possibly inaccurate, rainfall data. The reconstructed precipitation is then used to infer the hydrological model parameters, which are thus protected from the deteriorating effect of the uncertainty on the rainfall observations. This approach considerably reduces the bias in the inferred parameters and therefore leads to more reliable runoff predictions, which can be in turn very useful for decision makers in planning and policy making.

The use of AD makes the algorithm in principle applicable to more complex models. Indeed, the generalization of the algorithm from the toy model used in Albert et al. (2016) presented only one possibly challenging requirement, namely rewriting the Hamiltonian $\mathcal{H}_{\mathrm{HMC}}$, while the rest of the implementation of the algorithm remained essentially unchanged. This application shows that $\approx 10^3$ parameters can be inferred in less than 2 hours. This leaves us with a considerable margin to tackle more complex problems and datasets of much larger sizes. However, it must be noted that the HMC algorithm in the form described in this work is limited to non-spatially resolved models depending only on a time variable. In principle, the method could cope with further spatial dimensions, e.g., describing inundations, but this would require a non-trivial adaptation of the algorithm, which is not discussed here. More importantly, a significant limitation of the HMC method presented in this work is that it requires, unlike methods that only involve forward-solving the model, such as SIP (Del Giudice et al., 2016), an explicitly discretized solution of the hydrological model (see Eq. 5), which might not be always easily available. In those cases, one may need to resort to appropriate numerical solvers, often employing advanced implicit schemes, which in turn may make AD problematic.

The extension of the HMC method described here to further hydrological models and systems will be the focus of future works. Furthermore, the HMC algorithm presented here is not at all limited to hydrology. It is a very general, efficient, easily parallelizable and scalable algorithm that makes Bayesian inference with expensive stochastic models feasible in spite of its computational hardships, with a very broad range of potential applications in applied sciences that can benefit from stochastic modelling and a fast Bayesian inference method.

*Code and data availability.* The C++ code for HMC and the data used for this study are available on Github at https://github.com/ulzegasi/HMC_SIP.git.

**Appendix A**

We shall first describe the general procedure to derive the prior probability density $f(\xi)$ for a generic nonlinear SDE model, hence examine the specific case of the model used in this study (Eq. 2).

Let us consider a SDE model in the form,

$$\dot{\xi} = F(\xi,t) + G(\xi,t)\eta(t)\,, \tag{A1}$$

interpreted according to the Itô convention, where $\eta$ indicates white noise, with $\langle \eta(t)\eta(t')\rangle = \delta(t-t')$, and $F$ and $G$ are generic, possibly nonlinear, functions. The probability density for a full model realization can be written as the probability density for the corresponding noise realization $\eta$, that is,

$$f(\eta) \propto \exp\left[-\frac{1}{2}\int_0^T \eta^2(t)dt\right]\,, \tag{A2}$$

where $[0,T]$ is the observation time interval. Then, the probability density $f(\xi)$ is easily obtained by changing coordinates from the noise $\eta$ to the state variable of interest $\xi$, using the model of Eq. A1, that is,

$$\eta(t) = \frac{\dot{\xi}(t) - F(\xi,t)}{G(\xi,t)}\,. \tag{A3}$$

Depending on the SDE convention, this transformation might generate additional terms stemming from the Jacobian $d\xi/d\eta$ (Lau
and Lubensky, 2007). Finally, it is convenient to write the probability density $f(\xi)$ in terms of an action $\mathcal{S}(\xi,\dot{\xi})$ as,

$$f(\xi) \propto \exp\left[-\mathcal{S}(\xi,\dot{\xi})\right]\,. \tag{A4}$$

    Let us consider now the specific case of the stochastic process of Eq. 2. Using the procedure described above, the action $\mathcal{S}$ can be written in its continuous form (Lau and Lubensky, 2007) as,

$$\mathcal{S}(\xi,\dot{\xi}) = \int_0^T \frac{\tau}{4}\left(\dot{\xi}(t) + \frac{\xi(t)}{\tau}\right)^2 dt\,, \tag{A5}$$

where $T$ is the total measurement time. Following Albert et al. (2016), we rewrite Eq. A5 in the form,

$$\mathcal{S}(\xi,\dot{\xi}) = \int_0^T \left(\frac{\tau}{4}\dot{\xi}^2 + \frac{\xi^2}{4\tau} + \dot{\xi}\frac{\partial U(\xi,t)}{\partial\xi}\right) dt\,, \tag{A6}$$

with $U(\xi,t) = \xi^2(t)/4$. Using $\dot{\xi}\frac{\partial U}{\partial\xi} = \frac{dU}{dt} - \frac{\partial U}{\partial t}$ and $\frac{\partial U}{\partial t} = 0$, it is straightforward to obtain,

$$\mathcal{S}(\xi,\dot{\xi}) = \frac{\xi^2(T)}{4} - \frac{\xi^2(0)}{4} + \int_0^T \left(\frac{\tau}{4}\dot{\xi}^2 + \frac{\xi^2}{4\tau}\right) dt\,, \tag{A7}$$

which yields the discretized expression of Eq. 16.

## Appendix B

The first harmonic term of the discretized action (Eq. 16) can be written in the form,

$$\sum_{i=2}^{N} \frac{\tau}{4dt} (\xi_i - \xi_{i-1})^2 = \frac{\tau}{4} \sum_{s=1}^{n_P} \left[ \frac{(\xi_{(s-1)j_P+1} - \xi_{sj_P+1})^2}{j_P dt} + \sum_{k=2}^{j_P} \frac{k}{(k-1)dt} (\xi_{(s-1)j_P+k} - \xi_{(s-1)j_P+k}^*)^2 \right], \tag{B1}$$

with $\xi_{(s-1)j_P+k}^*$ defined by Eq. 19. Using the coordinate transformations of Eqs. 17 and 18, it is straightforward to rewrite B1 as,

$$\sum_{i=2}^{N} \frac{\tau}{4dt} (\xi_i - \xi_{i-1})^2 = \frac{\tau}{4j_P dt} \sum_{s=1}^{n_P} (u_{(s-1)j_P+1} - u_{sj_P+1})^2 + \frac{k\tau}{4(k-1)dt} \sum_{s=1}^{n_P} \sum_{k=2}^{j_P} u_{(s-1)j_P+k}^2. \tag{B2}$$

Moreover, using Eqs. 17 and 20 one obtains,

$$\sum_{i=2}^{N} \xi_i^2 = \sum_{s=1}^{n_P} \left[ u_{sj_P+1}^2 + \sum_{k=2}^{j_P} \left( \sum_{l=k}^{j_P+1} \frac{k-1}{l-1} u_{(s-1)j_P+l} + \frac{j_P-k+1}{j_P} u_{(s-1)j_P+1} \right)^2 \right]. \tag{B3}$$

Finally, using B2 and B3, the action $\mathcal{S}(\boldsymbol{\xi})$ (Eq. 16) can be formulated in the space of $u$-coordinates as in Eq.21.

## Appendix C

The runoff observation model $f\left(\boldsymbol{Q}_{\mathrm{obs}} \mid \boldsymbol{\xi}, \boldsymbol{\theta}, \sigma_z\right)$ of Eq. 8 needs to be formulated in the $u$- rather than $\xi$-space. The only term affected by this coordinate transformation is obviously the $\xi$-dependent term $Q_{M,(s-1)j_Q+1}^{(\xi)}$ in the model predicted discharge $Q_{M,(s-1)j_Q+1}(\boldsymbol{\xi}, \boldsymbol{\theta})$, with $s = 1, ..., n_Q + 1$ (see Eqs. 6 and 7). Using the definition of Eq. 7, straightforward calculations yield the recursive form,

$$Q_{M,(s-1)j_Q+1}^{(\xi)} = \left(1 - \frac{dt}{K}\right)^{j_Q} Q_{M,(s-2)j_Q+1}^{(\xi)} + \left(1 - \frac{dt}{K}\right)^{j_Q-1} r\left(\xi_{(s-2)j_Q+1}\right) + \sum_{k=2}^{j_Q} \left(1 - \frac{dt}{K}\right)^{j_Q-k} r\left(\xi_{(s-2)j_Q+k}\right), \tag{C1}$$

and then with the transformations of Eqs. 17 and 20, after replacing the number of discretization bins for the precipitation data $j_P$ with that for the discharge data $j_Q$,

$$Q_{M,(s-1)j_Q+1}^{(u)} = \left(1 - \frac{dt}{K}\right)^{j_Q} Q_{M,(s-2)j_Q+1}^{(u)} + \left(1 - \frac{dt}{K}\right)^{j_Q-1} r\left(u_{(s-2)j_Q+1}\right)$$
$$+ \sum_{k=2}^{j_Q} \left(1 - \frac{dt}{K}\right)^{j_Q-k} r\left(\sum_{l=k}^{j_Q+1} \frac{k-1}{l-1} u_{(s-2)j_Q+l} + \frac{j_P-k+1}{j_P} u_{(s-2)j_Q+1}\right), \tag{C2}$$

with $s = 2, ..., n_Q + 1$ and $Q_{M,1}^{(u)} = 0$. Finally, one has,

$$Q_{M,(s-1)j_Q+1}(\boldsymbol{u}, \boldsymbol{\theta}) = \frac{S_1}{K} \left(1 - \frac{dt}{K}\right)^{(s-1)j_Q} + A\frac{dt}{K} Q_{M,(s-1)j_Q+1}^{(u)} + \left[1 - \left(1 - \frac{dt}{K}\right)^{(s-1)j_Q}\right] Q_{\mathrm{gw}}. \tag{C3}$$

*Author contributions.* CA conceived the original idea, SU and CA designed the overall study and developed the theory. SU developed the code and performed the simulations. SU prepared the manuscript with contributions from CA.

*Competing interests.* The authors declare that they have no conflict of interest.

*Acknowledgements.* We wish to thank the High Performance Computing team of the Zurich University of Applied Sciences in Wädenswil, Switzerland, for the computational support. We are also grateful to Jörg Rieckermann, Eawag, for the permission to re-use the data from his group, originally published in Del Giudice et al. (2016), and to MeteoSwiss for the rainfall data for Sc1, re-used from the same publication. This work was supported by the Swiss National Science Foundation (Grant No 200021_169295).

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
