# Peer review of "Bayesian parameter inference in hydrological modelling using a Hamiltonian Monte Carlo approach with a stochastic rain model"

_EGUsphere, 2022_

## Author Response (AR1)

In the following, we present our responses to the referees.
Referees' comments are in blue.

**Referee #1**

**General comments**

The paper presents a Bayesian framework for forward and inverse problems in stochastic rain models based on time series observations of rainfall-runoff. The focus of the paper is on HMC as a scalable inference method. The paper provides a detailed study of the hydrological problem, giving a detailed description of the model, the parameters, and the priors. The discussion of the results is convincing.

We are grateful for the positive comments and the very interesting remarks and questions, which give us the opportunity to clarify some important points. We answer the specific comments one by one below.

**Specific comments**

1. It would be good to explicitly list the contributions, making it easier for the reader to see the high-level differences between this paper and Albert et al. (2016).

   In Albert et al. (2016) we described a novel implementation of an HMC algorithm combined with a multiple time-scale integration for Bayesian parameter inference with nonlinear stochastic differential equation models, and we demonstrated the performance of the method using a simple rainfall-runoff toy model and synthetic data time-series. For purely didactic purposes, the rain input to the system was modeled using a smooth sinusoidal function.

   In the present work instead, we apply the HMC method with the time-scale separation approach from Albert et al. (2016) for the first time to a real-world case study in urban hydrology, using real time-series of observed rainfall and outflows. Moreover, in this work we carry out the inference process using intentionally inaccurate rainfall observations and demonstrate the ability of the algorithm to reconstruct with great accuracy the unknown true average rainfall over the catchment. The reconstructed precipitation is then used to infer the hydrological model parameters, which are thus protected from the corrupting effect of the uncertainty on the rainfall observations.

   We have expanded section 5 ("Conclusions") to clarify the points above.

2. I wanted to clarify some of the results. Looking at Fig. 5-6 for predictions, it seems that the discharge data alone is mostly enough to provide good predictions for the rainfall. Is that generally true? How good would be the estimates of the parameters, and the predicted rainfall if you had no rain observation data? Would it be better or worse than the low-quality data of Sc2?

   This is indeed a very interesting point. We have run the HMC inference without rainfall data, obtaining both model parameter marginals and a predicted rainfall pattern that are substantially identical to those obtained with the inaccurate data of Sc2. Therefore, in Sc2 the HMC algorithm "learns" that the observed rain should be essentially ignored, thus producing results which are practically the same as if the inference was run without any rain data at all. However, in most applications the accuracy and reliability of the measured precipitation data is unknown

a priori. We show here that in those cases the rainfall observations can be safely used in the inference process, since the algorithm itself will assess its accuracy and possibly disregard it in favor of a more reliable reconstructed rainfall.

We believe that this result is interesting and worth a remark. Therefore, we have added it at the end of section 4.2 ("Results").

3.  In Fig 3, you show that with a more accurate dataset (Sc1) the estimates of the parameters are more sharply peaked (less uncertain), which makes sense. There seems to be a mismatch for some of these parameters, e.g. lambda and gamma. I guess the problem of inferring gamma and lambda jointly is ill-posed as they both define the transformation. If so, that could be an interesting point to discuss.

    This is also an interesting point. The inferred posterior distribution does not show any correlations between the parameters, e.g., lambda and gamma. Therefore, the problem of inferring them does not seem to be ill-posed.

    Instead, in Sc1 the HMC algorithm tunes the parameters of the rainfall potential transformation to match the (accurate) rainfall data. This is evident for the large precipitation peak near time = 60 min, as clearly visible in the lower halves of figures 5 and 6. The smaller value of the inferred parameter gamma in Sc1 reflects exactly this attempt of the algorithm to find a better fit to the rain observations, especially where precipitation values are large. The smaller observational error for the precipitation in Sc1 is also an obvious consequence of a better match of predictions and data. All other parameter marginals exhibit much smaller discrepancies between Sc1 and Sc2.

    This point is now clearly addressed in the revised version of the manuscript (section 4.2, "Results").

4.  I'm also generally curious what is the end goal of this study, for example, can these results be used to aid policy making? Are quantities like groundwater flow, or retention time important to know for planning purposes? Would there ever be a need to run a system like this in real-time?

    This work is intended to be a purely methodological study. Its main goal is to demonstrate that the HMC algorithm combined with a multiple time-scale integration presented in Albert et al. (2016) can be successfully applied to solve real-world hydrological inference problems with computationally expensive stochastic models. This method is especially very well-suited for cases, far from rare in hydrology, where the precipitation data is inaccurate and unreliable. It reduces considerably the bias in the inferred parameters by shielding them from the deteriorating effect of the rainfall data inaccuracy, thus leading to more reliable runoff predictions. The knowledge of all model parameters, including the groundwater flow and the retention time, is essential for making robust probabilistic predictions, which can certainly be useful in planning and policy making. This method is definitely a powerful and versatile tool for Bayesian inference with expensive stochastic models, whereas it might not be the optimal solution for real-time control of hydrological systems, where faster algorithms might be

preferable. This topic, however, is not discussed in detail here since it goes beyond the scope of this study. We have added a short remark in section 1, "Introduction".

**Technical comments**

1. Some references need fixing, e.g. Line 510, some papers are missing titles.

2. Line 290: construct e reversible -> construct a reversible

We thank the reviewer for pointing out these two technical issues, which have been fixed in the revised manuscript.

**Referee #2**

**General comments**

This manuscript presents a Bayesian framework for forward and inverse problems and presents the Hamiltonian Monte Carlo (HMC) as a scalable inference method for calibration of models to noisy time series. The paper details an application of this framework in stochastic rain models based on time series observations of rainfall-runoff. The paper provides a case study of a single storm event over a single catchment, and although the paper is technically well written, it currently reads more like a technical note rather than a research article. Overall, the implications of the study were unclear – floods are mentioned briefly but discussion of whether this approach holds up when considering 1. different hydrological modelling approaches, 2. climate variability and non-stationarity, 3. different catchment types and antecedent conditions, and 4. flash floods, could further strengthen the argument for using this novel approach.

This paper is more akin to a technical report and is therefore not entirely suited for HESS audiences as a research article in its current form. However, the approach detailed in the paper and its suitability to model real world hydrological impacts are of interest to HESS audiences. Thus, the manuscript could be strengthened with some **moderate revisions** and reframing; to demonstrate the superiority of this methodology and approach, where the application of such an approach is most beneficial, and, what the implications of using this approach are in terms of hydrological services to aid decision makers. The inclusion of the above would go most of the way to addressing "relevant scientific questions within the scope of HESS" as well as providing more tangible implications for the reader.

Unfortunately, as a reviewer I only have an option to choose between minor and major revisions so I chose major to reflect the fact that the effort required to address my comments would be greater than that of addressing minor comments. However, I suspect that the effort needed to revise this manuscript would fall somewhere in between – i.e. moderate revisions. I sincerely hope these comments and the more specific ones below are helpful to the authors.

We are grateful to Referee #2 for the helpful remarks and questions. We agree that some aspects of our work needed to be clarified and therefore we have added further comments and explanations in the revised manuscript.

Here, before we set off to answer the specific comments, we would like to make some general remarks. This work is indeed intended to be a methodological article with emphasis on a powerful and versatile inference method for stochastic models rather than on specific hydrological problems and models. Nevertheless, we believe that this study is of great interest for the hydrological community and therefore for the HESS audiences. Stochastic models in hydrology are very useful and widespread tools for making reliable probabilistic predictions. However, such models can be used for making predictions only if model parameters are first of all calibrated to measured data in a consistent framework such as the Bayesian one. Unfortunately, Bayesian parameter calibration, a.k.a. inference, with stochastic models, turns out to be an often computationally intractable problem. Therefore, the computational obstacle is often overcome by employing over-simplified error models, which lead to biased parameter estimates and unreliable predictions. Our goal in this work is to present a HMC algorithm that makes Bayesian parameter inference with stochastic models possible, from which hydrology can potentially take great advantages: a sound calibration of model parameters is essential for making robust probabilistic predictions, which can certainly be useful in planning and policy making. Discussing

specific hydrological models or systems is outside the scope of our present work, and will be the focus of further studies.

**Specific comments**

1. I was not sure what the benefits of this approach are versus other methods. For example, why is this approach is beneficial over other hydrological modelling approaches, such as hydraulic or other physics-based approaches (e.g. flow routing) or conceptual models (e.g. pipe flow simulations) in terms of computational efficiency and accuracy. For example, could you run this simulation in real time for now casting?

   This is a very interesting question that gives us the opportunity to clarify an important point. In this work we do not propose a new hydrological modeling approach, but rather a method for the calibration (i.e., inference) of the model parameters of computationally expensive stochastic models. Stochastic models are becoming very popular in hydrology as they allow a more realistic description of real systems by incorporating the noise directly into the model. In particular, input errors associated with an inaccurate knowledge of the rainfall, which often represent a major source of uncertainty in hydrological modeling, can be integrated in the model in the form of a stochastic input process. Moreover, Bayesian inference methods bear the great advantage over traditional optimization algorithms of providing an uncertainty estimation for the calibrated parameters in the form of a probability distribution. The knowledge of such uncertainty is of paramount importance for making probabilistic predictions, which can be in turn a very important aid to decision makers. The HMC method presented in this work is a fast, efficient, parallelizable and scalable algorithm that makes Bayesian inference with stochastic models feasible in spite of its computational hardships. However, it might not be the best option for real-time control of hydrological systems, where faster algorithms should be preferable. This latter point is not discussed in detail since it goes beyond the scope of this study.

   We have added a comment in section 1, "Introduction".

2. What catchment conditions is this method suitable for? I assume modelling storm water is the reason for choosing an urban catchment for a case study, but perhaps the paper could be strengthened by stating that explicitly and focusing on the difficulty of modeling storm water runoff accurately.

   The case study presented here was chosen to put emphasis on input uncertainties, which often lead to biased parameters, badly calibrated models and unreliable predictions. Precipitation is often the biggest source of uncertainty in hydrological modeling in general, not just in urban catchments. Therefore, the HMC method is by no means limited to urban hydrology, and could certainly be applied in natural catchment hydrology as well. The latter, however, goes beyond the scope of this work, since our focus is on parameter inference with a stochastic input process. More information on possible hydrological applications can be found in Del Giudice et al., Water Resour. Res., 52, 2016.

   This is discussed more clearly in section 1, "Introduction", of the revised text.

3. What applications is this methodology suited for? Floods are briefly mentioned and storm water is the focus, but does this methodology enhance the modelled accuracy of any other flood impacts such as inundation?

The HMC algorithm presented in this paper is a powerful method for high-dimensional inference problems, where not only a few model parameters are calibrated, but the full rainfall pattern is also reconstructed with great accuracy. However, the HMC algorithm in the form described in the paper is limited to non-spatially resolved models depending only on a time variable. In principle, the method could cope with further spatial dimensions (e.g., inundations), but this would require a non-trivial adaptation of the algorithm, which is not discussed here. In other words, our goal is to show that high-dimensional Bayesian inference with stochastic hydrological models is possible in spite of its computational difficulty. However, how far one can go in terms of additional dimensions remains an open question, which will be the subject of further studies.

We have clarified this point in section 5 of the revised manuscript ("Conclusions").

4. The methods section takes up the bulk of the paper, even allowing for the fact that the focus is on the novelty of the method. Could some of the details be put into supplementary information? It is unclear as to how novel the methodology or framework proposed is given that a prior paper on the HMC has been published. Could the authors please highlight what is "new"?

The method is the main focus of this work, which is the reason why it takes up most of the paper. We believe that the details presented in the text are important for the reader to grasp the subtleties that make HMC a powerful and efficient inference method for stochastic models. It is important to underline that the HMC method itself was first proposed in the 1980s and it is thus not new at all. In the present work, we apply an HMC method with a novel time-scale separation approach (from Albert et al., 2016) for the first time to a real-world hydrological case study, using real time-series of observed rainfall and outflows. Moreover, we also demonstrate for the first time the ability of the algorithm to reconstruct with great accuracy the unknown true average rainfall over the catchment using only prior knowledge and the observed outflow. The reconstructed precipitation is then used to calibrate the hydrological model parameters, which are thus protected against the degrading effect of the possible rainfall data inaccuracy. This considerably reduces the bias in the inferred parameters, thus leading to more realistic models and reliable runoff predictions.

We have modified section 5 ("Conclusions") to address more clearly the novelty of this work.

5. The case study description is a bit light on in detail. I could not discern the reasons why the single storm event and catchment were chosen from the case study description. A reader is likely to be skeptical as to the broad applicability of any method when only one catchment and event are modelled, can the case study be expanded to include multiple events and/or multiple catchments? In addition, the reasons for the two ScX datasets could be made clearer to the reader in the case study description. Also is there a third case that could be explored? No Sc1 or Sc2 data?

One can find more details on the case study and a more comprehensive analysis based on multiple independent time-series in Del Giudice et al., Water Resour. Res., 52, 2016. Here, the goal is to introduce the HMC method and demonstrate its performance with the computationally hard task of inferring simultaneously both the model parameters and the true average rainfall over the catchment using real-world observations. The single event considered in the paper represents a worst-case scenario due to the large inaccuracy of the rainfall data, and therefore a more challenging and probative test ground for the inference method. Since the focus of this paper is purely on the methodology and not the exploration of different hydrological models and systems, we are planning to consider other possible applications in future studies.

We have expanded section 4 ("Case study") to address the above points.

6. Figure 3 could be further or more thoroughly explained in terms of the whys – e.g. why does gamma show the largest shift? Is it due to, for example, the event characteristics or catchment characteristics or both?

This is another very good point. Essentially, the algorithm tunes the model parameters to match the different rainfall data of Sc1 and Sc2. So, for instance, the smaller value of the inferred parameter gamma in Sc1 reflects exactly this attempt of the algorithm to find a better fit to the rain observations, especially at time points where precipitation values are large, such as the precipitation peak near time = 60 min clearly visible in the lower halves of figures 5 and 6.

We have expanded our discussion of Figure 3 to clarify this important aspect (section 4.2, "Results").

7. The comparison between poor quality and good quality rainfall is a bit confusing (Figures 5-6) but looks like an interesting result? It appears that the discharge data alone is enough to provide good predictions for the rainfall. What is the purpose of using Sc2 then – can the authors please explain this in detail? Also, it would be good to know whether this is the case across the board (i.e. more than one event in one catchment).

This is indeed a fundamental point of our work. We have run the HMC inference without rainfall data, obtaining both model parameter marginals and a predicted rainfall pattern that are substantially identical to those obtained with the inaccurate data of Sc2. Therefore, in Sc2 the HMC algorithm "learns" that the observed rain should be essentially ignored, thus producing results which are practically the same as if the inference was run without any rain data at all. However, in most applications the accuracy and reliability of the measured precipitation data is unknown a priori. We show here that in those cases the rainfall observations can be safely used in the inference process, since the algorithm itself will assess its accuracy and possibly disregard it in favor of a more reliable reconstructed rainfall. This is a very important result, which is explicitly discussed in the revised manuscript at the end of section 4.2, "Results".

8. Results overall: A discussion of the limitations and applicability/suitability of the method along with the implications of its use would strengthen the paper. The figure discussions could be improved by relating the results to the characteristics of the event and catchment. Could the

authors please detail a real-world application using this approach (e.g. nowcasting of storm water runoff during an event)?

As already mentioned above, we believe that the HMC method presented here is in general not optimal for real-time control applications. Instead, the HMC algorithm is an inference method for the data-driven calibration of hydrological model parameters, especially very well-suited for 1) computationally expensive stochastic models, and 2) cases, far from rare in hydrology, where the precipitation data is inaccurate and unreliable. It considerably reduces the bias in the inferred parameters by shielding them from the deteriorating effect of the rain inaccuracy, thus leading to more reliable runoff predictions based on rainfall predictions. The section "Conclusions" of the revised text has been expanded to better clarify these points.

9.  In considering the citations and reference list, it appears to me that the authors have considered all the major technical HMC and related publications (although I am far from a leading expert in the field of HMC), however I note here, that in addressing the above and general comments, more citations for the background and contextual information will need to be included.

We have updated our references in the introduction. More references to the hydrological literature can also be found in Del Giudice et al., Water Resour. Res., 52, 2016.

**Technical comments**

Given that the paper could be improved and strengthened by reframing and providing more context for the reader, and thus would need moderate revisions, I have not gone through the manuscript with a fine-tooth comb, however I have picked a couple of things up:

References: Some references need fixing, for example, some papers are missing titles.

Edits: Line 290: construct e reversible should be "construct a reversible"

Finally, we thank the reviewer for pointing out a couple of technical issues, which have been fixed.

**Referee #3**

**General comments**

The paper deals with the calibration of a hydrological process model subject to stochastic input. The calibration is performed using Bayesian inference with two different data sets that use the same runoff time series. However, the first data set relies on high-resolution rainfall data, while the rainfall data in the second case are marred by low correlation with the output and lower temporal resolution. To successfully accomplish calibration, the authors describe and use a method a called Hamiltonian Monte Carlo, which uses the posterior density as the potential energy of a dynamic system that is integrated in auxiliary time. Although the hydrological model is fairly simple, the paper is interesting to the broader community as it uses a very powerful (but fairly unknown) inference approach to solve a difficult task of widespread interest. Additionally, the paper is very well-written. I recommend publication after the major concerns described below are addressed.

We thank Referee #3 for the interesting comments, and we are glad to address the specific issues one by one below.

**Specific issues**

**1**. Throughout the manuscript the authors state that the approach is very general and easy to port to different use cases. For example, from the Introduction:

"*In Albert et al. (2016), we claimed that the HMC algorithm, combined with the multiple time-scale integration, would be applicable to a wide range of inference problems [...]. We show that with only a little analytical effort the HMC method can be extended from the toy model and the smooth synthetic data used in Albert et al. (2016) to a real-world hydrological case study with real noisy rainfall and runoff time series. [...]. Indeed, the HMC method is by no means limited to an OU process, unlike the original SIP approach of Del Giudice et al. (2016) [...]. Although in this study we also opt for an OU process for the sake of simplicity, it should be clear that such process could be arbitrarily replaced by any other stochastic process [...].*"

By reading the paper, I am not convinced that the method is as simple, easy to generalize, apply, and scale as claimed. It worries me that the "little analytical effort" for a relatively simple case-study like this one amounts to about 20 equations in the main text plus three appendices, and still falls short, as there is not an explicit working-out of eq. (6) from (5). Hence, my questions:

- Is it always granted that the "analytical steps" are doable?

- Given that in Appendix A the definition of the Langevin dynamics is explicitly used for the derivation, what would happen with a stochastic process more complex than Langevin?

Given the above, here are my suggested minimal actions required for publication:

- Please, substitute "little analytical effort", with "substantial analytical effort" or similar wording.

- In the Introduction, please, make it clear that the requirement to integrate the Hamiltonian dynamics implies the need for a full analytical specification of the model. This includes being explicit about the form of $f(\xi)$, which might be difficult for more complex cases. Please, be also explicit that this is a characteristic (a drawback really) of HMC, as with more traditional

sampling techniques one would just sample from $f(\xi)$, without the need for calculating the relevant action. Please, be also upfront about the physical meaning of the action, which is a bit obscure, and whether there are any limitations attached.

- Please, add a Discussion section (or embed it in the Conclusion) to discuss the analytical requirements, efforts, and foreseeable limitations for different use cases.

Several very interesting points are raised here, which certainly need clarification in the next revision of the manuscript. It is definitely true that the integration of the Hamiltonian dynamics requires both a full analytical specification of the model and an explicit form for f($\xi$). It is also true, however, that the HMC method presented here is applicable to generic stochastic models. Indeed, writing an explicit form for f($\xi$) is not actually difficult and always doable, even for more complex cases. Consider for instance a SDE model in the generic form (upon discretization):

$\xi_{i+1} = F(\xi_i) + G(\xi_i)\, \eta_i$

with noise $\eta$. The analytical steps to obtain f($\xi$) essentially consist in writing the density for a full model realization as the product of the probabilities of the individual $\eta_i$, that is,

$f(\eta_1, \eta_2, ...) = \prod_i f(\eta_i)$

then change coordinates from to the state variable of interest, in this case $\xi$, using the model above, i.e.,

$\eta_i = (\xi_{i+1} - F(\xi_i))/G(\xi_i),$

without forgetting the Jacobian d$\xi$/d$\eta$. In the specific case presented in this work, although we use a linear SDE model for the rainfall potential , we actually never exploit its linearity. The derivation of the density f($\xi$) is thus by no means limited to linear SDE models.

It should be noted, however, that while f($\xi$) does not represent a major issue, the actual limitation of the HMC method lies rather in the hydrological ODE model, which we discretize using an explicit forward scheme, which imposes strict limitations on the time step to guarantee numerical stability. As explained in the text, in regions where the solution shows a rapidly varying behavior, an implicit method would be numerically more stable, although more difficult to implement. Moreover, in the case of more complex models, when an explicit discretized version is not readily available, one might need to resort to appropriate numerical ODE solvers, often employing advanced implicit schemes, which may make automated differentiation problematic.

We agree that we should discuss such limitations more clearly in the next revision of the paper.

Moreover, we will also be more upfront about the physical meaning of the action and state explicitly that it is simply the negative log of the probability density f($\xi$). This is a particularly interesting point. Indeed, in this regard, traditional sampling techniques require the same analytical effort as the HMC method since they need to know the probability density f($\xi$), which is essentially the same as the action. Instead, we would like to point out more clearly that the main characteristic (or drawback) of the HMC method as described in this work, compared to other sampling techniques, is that it requires an explicit discretized version of the hydrological ODE model. Such analytical expressions might not be always easily available.

In the revised text, we have replaced expressions like "little analytical effort", "easily scalable", "effortless" with more appropriate expressions. Moreover, we have expanded section 2.5 ("The priors"), Appendix A and section 5 ("Conclusions") to address all the points above.

As somewhat minor issues, but still quite important, I would like to point out what follows.

**2**. Given the description of the time discretization, that is: "*For this purpose, we subdivide each interval between consecutive rain observations into Jp bins […]. The number of discretization points is thus the same (N) in both rainfall and runoff dimensions, and it defines the discretization time step dt=T/(N−1), where T is the total time interval covered by observations*"

- If the observations were irregularly spaced in time, would different observation windows have finer/coarser time discretization as Jp is constant? What would this imply?

- Is it required to have the same discretization points for rainfall and runoff?

I suggest not to miss the opportunity to shed light on the above questions explicitly.

These are certainly interesting as well as important implementation details, which we are very glad to clarify. The only requirement of the method is that the total number N of discretization points is large enough compared to the number of measurement points in order to accurately probe the fine dynamics occurring on short time scales between observations. Other features described in this paper, such as for instance having the same number of discretization points for rainfall and runoff, are just arbitrary choices to ease the practical implementation of the method, which could be removed without altering the results and conclusions of our work. The same holds if the observations were irregularly spaced in time, in which case one could use observation windows with more/less intermediate discretization points. In other words, using equally spaced observations and the same discretization points for rainfall and runoff simplify some implementation details, which we did not deem essential in order to demonstrate the applicability of the method and its potential benefits. Note that increasing the number of discretization points (N), with a fixed number of observation points, does only moderately increase the computational effort, since the part of the Hamiltonian dynamics that scales with N can be calculated analytically. We have explicitly addressed the points above in section 2 ("Bayesian inference with a stochastic rain model") of the revised text.

**3**. The calculations to go from eq. (5) to eq. (6) should be included in an Appendix.

The calculations to go from eq. (5) to (6) are actually straightforward and can be easily included in the main text, rather than in a dedicated appendix.

**4**. Given the sentence "*Our prior knowledge of the rainfall potential ξ is defined in terms of a function S(ξ), called action (Lau and Lubensky,2007; Albert et al., 2016) […],*":

- How general is the definition of action?

- Does it depend on the stochastic process at hand? Does eq. (16) depend on the chosen stochastic process?

I recommend that the authors discuss the above questions explicitly in the manuscript.

We will gladly address these questions explicitly. The action is by definition just the negative log of the probability density of the process and as such it depends on the specific stochastic process at hand. Therefore, eq. (16) depends on the stochastic process of eq. (2). However, we would like to remark again that calculating the action, given a stochastic process, is in general a relatively simple task. This is explained in the revised text in both section 2.5 and appendix A.

**5**. By reading "*Before setting off to implement the HMC algorithm, we need to take one further fundamental step, i.e., we apply the transformation from the coordinates ξ to the so-called staging variables u*", I should ask:

- It seems a transformation to canonical coordinates, but besides this guess, it is unclear why this step that adds to the complexity of the overall strategy (eq. (17) - (21)) is essential.

- Is this transformation dependent on the choice of model, stochastic process, etc. made by the authors? How generalizable is this step?

I believe that the authors should not miss the opportunity to discuss the above questions explicitly.

We will certainly discuss these points in more detail. The Referee is actually right that the transformation ξ→u is analogous to a transformation to canonical coordinates, and it is also true that such transformation adds a further degree of complexity to the overall strategy. However, it also bears significant benefits. The transformation makes it possible to decouple dynamics occurring on very different time scales, and allows us to integrate analytically the fast component, analogous to a system of coupled harmonic oscillators, yielding a substantial computational speed-up. Therefore, although the coordinate transformation is not essential for the HMC method (which would work without it), it turns out to be computationally very convenient, and we consider its advantages worth the implementation effort.

Regarding the generalizability of this approach, as explained in Albert et al. (2016), the decoupling of the different dynamics is always possible for 1D SDE models. In this work we focus only on 1D models and we leave the exploration of higher-dimensional models to future studies.

These points are discussed in the revised manuscript at the end of sections 2.5 ("The priors") and 3 ("The HMC algorithm").

**6**. It is unclear to this reviewer what equations (22) and (23) add, as the posterior *f* is explicit on both sides of the equation.

The purpose of eq. (23) is to show how auxiliary degrees of freedom, i.e., the momenta π and p, are added to the system. The HMC method samples this higher-dimensional space. Then, eq. (22) shows how the auxiliary momenta are integrated out, thus yielding the sought posterior. This is in essence how HMC methods work.

**7**. From "*The HMC algorithm iterates the following steps. First, vectors of momenta π and p are drawn from the normal distributions defined by the kinetic terms in Eq. 23.*" it is unclear how eq. (23) defines the kinetic energy.

- What's the "effective temperature of the system"?

- In a complex scenario, how is the magnitude of the momenta chosen to make sure that all the local minima of the Hamiltonian can be escaped while assuring that relatively flat areas are not overlooked?

Please, discuss these points in the paper.

This is an important point. The Hamiltonian of eq. (23) contains a "potential energy" term, that is, the negative log of the posterior f, and two additional terms depending on the auxiliary momenta π and p,

akin to the kinetic energies associated with the degrees of freedom of a fictitious statistical mechanics system. The HMC method describes such system dynamics through the integration of Hamilton equations. We do not control these Hamiltonian dynamics with an effective temperature, but rather by tuning the fictitious masses in eq.(23), which represent the variances of the normal distributions for the corresponding momenta. Therefore, small/large masses correspond to small/large momenta. This is described in more detail in Albert et al. (2016). Moreover, in this work we opt for a manual tuning of the masses for ease of implementation, although more advanced and automated methods are available, as explained in section 4.1. The presence of pronounced local minima might present an insurmountable obstacle, even for refined HMC methods and necessitate further enhancements such as Metadynamics. However, they do not seem to be an issue here.

This is discussed explicitly text in section 3 ("The HMC algorithm") of the revised text, after eq. (23).

**8**. Referring to the sentence "*Using the definitions of Sections 2.1, 2.2, 2.3, 2.4 and 2.5, we write the HMC Hamiltonian as [...].*", please add "*eq. (1)*" to the list of what has been used.

Done.

**9**. Referring to "*In particular, we exploit the fact that the much faster dynamics of the intermediate discretization points described by HN is analogous to a system of uncoupled harmonic oscillators that can be solved analytically. This analytical solution gives a significant boosting contribution to the intrinsic efficiency of the HMC algorithm.*":

- How general is this statement with respect to the assumptions made in this study, e.g., chosen stochastic process, hydrological model, observational error models, etc.?
- Is it granted that such an "analytical boost" can always be enjoyed by any SDE model?

Please, discuss these points in the paper.

The statement is always true for 1-dimensional SDE models, such as the toy model of Albert et al. (2016) or the case study presented in this work. It also holds true in the case of multiple independent variables, where the decoupling procedure can be applied to each of them individually. In this way, the approach covers a significant range of possible hydrological modeling scenarios. This is clarified at the end of section 3.

**10**. Figure 6. - Can the y-axis be limited to the extent of Figure 5? This would ease comparisons.

Maybe we misunderstood the point raised by the Reviewer, but the y-axis of figures 5 and 6 are the same. In any case, figures 5 and 6 have been replaced in the revised manuscript by a single figure.

**11**. For comparisons, it could be useful to generate a Figure where P and Q densities from Sc1 and Sc2 are overlaid with transparency.

We agree that the comparison of figures 5 and 6 should be more straightforward. However, in order to avoid a single overcrowded figure, we prefer to place figures 5 and 6 side by side in one figure spanning two columns.

---

## Referee Report (RR1)

The authors have responded to all the questions and requests I made in the first round of reviews. Therefore, I recommend publication of this very interesting and well-written paper.

At the authors' discretion, I suggest that the paper could be further improved by addressing the following points:

- What is the mean value of the normal distributions used for drawing the momenta, eq. (23)? Is it unity?
- If the transformation (17)-(19) was non-linear, would the authors have had to set those equations forward in the problem statement, including eq. (11)-(13), i.e., the Jacobian?
- More than resorting to Meta-Dynamics, which is difficult to perform in high-dimensional spaces, shouldn't the author be more interested in parallel tempering? Wouldn't this also help in assessing what the effect of the specific values of the masses would be?
- I would be curious to know why 6 evaluations of the derivatives of the posterior are required in the molecular dynamics part. Is the derivation of the energy six times more expensive than its evaluation because these are 6 evaluations, or is it actually 6*6=36 times more expensive?

---

## Referee Report (RR2)

**General comments with respect to the 2$^{nd}$ review of the manuscript.**

The authors addressed all previous comments to my satisfaction; however I still would have liked to see another case study detailed to further support the proposition that the HMC could be used in a broad range of applications. Even so, I recommend accepting this manuscript with minor revisions. See my specific comments below.

**Specific comments with respect to the 2nd review of the manuscript**

**Specific comment 1: Abstract**

The authors addressed my general comments with:

Stochastic models in hydrology are very useful and widespread tools for making reliable probabilistic predictions. However, such models can be used for making predictions only if model parameters are first of all calibrated to measured data in a consistent framework such as the Bayesian one. Unfortunately, Bayesian parameter calibration, a.k.a. inference, with stochastic models, turns out to be an often computationally intractable problem. Therefore, the computational obstacle is often overcome by employing over-simplified error models, which lead to biased parameter estimates and unreliable predictions. Our goal in this work is to present a HMC algorithm that makes Bayesian parameter inference with stochastic models possible, from which hydrology can potentially take great advantages: a sound calibration of model parameters is essential for making robust probabilistic predictions, which can certainly be useful in planning and policy making. Discussing specific hydrological models or systems is outside the scope of our present work, and will be the focus of further studies.

This is a clearer, simpler version of the abstract and I suggest the authors use this instead. Something like:

Stochastic models in hydrology are very useful and widespread tools for making reliable probabilistic predictions. However, such models are only accurate at making predictions if model parameters are first of all calibrated to measured data in a consistent framework such as the Bayesian one. Unfortunately, Bayesian parameter calibration, a.k.a. inference, with stochastic models, is often a computationally intractable problem due to the expensive likelihood functions employed in traditional inference algorithms such as... . Therefore, the prohibitive computational cost is often overcome by employing over-simplified error models, which lead to biased parameter estimates and unreliable predictions. Our goal in this work is to present a very computationally efficient novel HMC algorithm which makes Bayesian parameter inference with stochastic models possible. We show that this approach is robust, by detailing a case study from urban hydrology which would normally be computationally prohibitive to calibrate. This work shows the potential of the approach; a sound calibration of model parameters is essential for making robust probabilistic predictions, which can certainly be useful in planning and policy making. Discussing specific hydrological models or systems is outside the scope of our present work and will be the focus of further studies.

**Specific comment 2: Introduction**

**In the introduction authors have addressed previous comments which is good but now it is more muddied to read, and it was already a bit clunky and hard to follow. Suggest restructuring to make clearer to the reader what the actual goal of the paper is and why the inference algorithm is novel.**

Suggested structure:

Cover concept of stochastic models in hydrology and their widespread use – what applications and why they are useful.

Cover concept of calibration aka inference in a Bayesian framework and why it is important for accuracy.

Introduce the concept of computationally prohibitive costs of traditional inference algorithms and what everyone else does – employ the use of over simplified error models and why this is bad.

Introduce novel inference HMC algorithm and why it is computationally efficient and accurate and for which applications it can be employed.

Introduce goal of the study – to prove the HMC is accurate, efficient and effective by employing in case study which is hard to model: urban hydrology and why it's hard (inaccurate and unrealiable rainfall) - could even introduce the concept of the "worst case scenario".

Then state implications – advantages in using this across a broad range of hydro applications.

**Specific comment 2: Methods – still appears to be very long and not sure all details need to be in there.**

In addressing my previous comments, the authors state:

In the present work, we apply an HMC method with a novel time-scale separation approach (from Albert et al., 2016) for the first time to a real-world hydrological case study, using real time-series of observed rainfall and outflows. Moreover, we also demonstrate for the first time the ability of the algorithm to reconstruct with great accuracy the unknown true average rainfall over the catchment using only prior knowledge and the observed outflow. The reconstructed precipitation is then used to calibrate the hydrological model parameters, which are thus protected against the degrading effect of the possible rainfall data inaccuracy.

Following on from this, it seems that the method has already been detailed in Albert et al., and thus only the additional information pertinent to the case study, e.g. priors, hydrological and rainfall models used etc., need to be detailed.

Perhaps there could be some movement of the HMC algorithm description to an appendix?

**Specific comment 3: Conclusion – could be made stronger to convince the reader of the applicability of the HMC as an inference algorithm for stochastic hydrological models.**

Authors note from addressing previous comments:

Our goal in this work is to present a HMC algorithm that makes Bayesian parameter inference with stochastic models possible, from which hydrology can potentially take great advantages: a sound calibration of model parameters is essential for making robust probabilistic predictions, which can certainly be useful in planning and policy making. Discussing specific hydrological models or systems is outside the scope of our present work, and will be the focus of further studies.

And:

In the present work, we apply an HMC method with a novel time-scale separation approach (from Albert et al., 2016) for the first time to a real-world hydrological case study, using real time-series of observed rainfall and outflows. Moreover, we also demonstrate for the first time the ability of the algorithm to reconstruct with great accuracy the unknown true average rainfall over the catchment using only prior knowledge and the observed outflow. The reconstructed precipitation is then used to calibrate the hydrological model parameters, which are thus protected against the degrading effect of the possible rainfall data inaccuracy. This considerably reduces the bias in the inferred parameters, thus leading to more realistic models and reliable runoff predictions.

Suggested structure:

Restate goal of paper and scope.

State novelty – first time to reconstruct...

State what was demonstrated and why biases in the inferred parameters were reduced.

State limitations of the study.

State future work.

State implications for hydrology and why broadscale use of the HMC is advantageous.

To include the above salient points.

**Specific comment 4: Improve writing. Topics are introduced across two paragraphs or more (e.g. in the introduction). Sentences are too long and too complicated throughout.**

Unfortunately, some of the key messages get lost in the unwieldy structure of the paragraphs and sentences. Suggest containing a topic per paragraph and to shorten sentences. Make sure key messages stand out – that they are simple, clear and succinct. The manuscript could be made more impactful by improving the writing.

---

## Author Response (AR2)

In the following, we present our responses to the referees.
Referees' comments are in blue.

**Referee #2 – Report #2**

**Specific comment 1: Abstract**

The authors addressed my general comments with:

Stochastic models in hydrology are very useful and widespread tools for making reliable probabilistic predictions. However, such models can be used for making predictions only if model parameters are first of all calibrated to measured data in a consistent framework such as the Bayesian one. Unfortunately, Bayesian parameter calibration, a.k.a. inference, with stochastic models, turns out to be an often computationally intractable problem. Therefore, the computational obstacle is often overcome by employing over-simplified error models, which lead to biased parameter estimates and unreliable predictions. Our goal in this work is to present a HMC algorithm that makes Bayesian parameter inference with stochastic models possible, from which hydrology can potentially take great advantages: a sound calibration of model parameters is essential for making robust probabilistic predictions, which can certainly be useful in planning and policy making. Discussing specific hydrological models or systems is outside the scope of our present work, and will be the focus of further studies.

This is a clearer, simpler version of the abstract and I suggest the authors use this instead. Something like:

Stochastic models in hydrology are very useful and widespread tools for making reliable probabilistic predictions. However, such models are only accurate at making predictions if model parameters are first of all calibrated to measured data in a consistent framework such as the Bayesian one. Unfortunately, Bayesian parameter calibration, a.k.a. inference, with stochastic models, is often a computationally intractable problem due to the expensive likelihood functions employed in traditional inference algorithms such as... . Therefore, the prohibitive computational cost is often overcome by employing over-simplified error models, which lead to biased parameter estimates and unreliable predictions. Our goal in this work is to present a very computationally efficient novel HMC algorithm which makes Bayesian parameter inference with stochastic models possible. We show that this approach is robust, by detailing a case study from urban hydrology which would normally be computationally prohibitive to calibrate. This work shows the potential of the approach; a sound calibration of model parameters is essential for making robust probabilistic predictions, which can certainly be useful in planning and policy making. Discussing specific hydrological models or systems is outside the scope of our present work and will be the focus of further studies.

We have replaced the original abstract with the version suggested by the referee.

**Specific comment 2: Introduction**

**In the introduction authors have addressed previous comments which is good but now it is more muddied to read, and it was already a bit clunky and hard to follow. Suggest restructuring to make clearer to the reader what the actual goal of the paper is and why the inference algorithm is novel.**

Suggested structure:

Cover concept of stochastic models in hydrology and their widespread use – what applications and why they are useful.

Cover concept of calibration aka inference in a Bayesian framework and why it is important for accuracy.

Introduce the concept of computationally prohibitive costs of traditional inference algorithms and what everyone else does – employ the use of over simplified error models and why this is bad.

Introduce novel inference HMC algorithm and why it is computationally efficient and accurate and for which applications it can be employed.

Introduce goal of the study – to prove the HMC is accurate, efficient and effective by employing in case study which is hard to model: urban hydrology and why it's hard (inaccurate and unrealiable rainfall) - could even introduce the concept of the "worst case scenario".

Then state implications – advantages in using this across a broad range of hydro applications.

We have restructured the introduction taking the referee's suggestion into account. We hope that this makes it clearer and easier to read.

**Specific comment 2: Methods – still appears to be very long and not sure all details need to be in there.**

In addressing my previous comments, the authors state:

In the present work, we apply an HMC method with a novel time-scale separation approach (from Albert et al., 2016) for the first time to a real-world hydrological case study, using real time-series of observed rainfall and outflows. Moreover, we also demonstrate for the first time the ability of the algorithm to reconstruct with great accuracy the unknown true average rainfall over the catchment using only prior knowledge and the observed outflow. The reconstructed precipitation is then used to calibrate the hydrological model parameters, which are thus protected against the degrading effect of the possible rainfall data inaccuracy.

Following on from this, it seems that the method has already been detailed in Albert et al., and thus only the additional information pertinent to the case study, e.g. priors, hydrological and rainfall models used etc., need to be detailed.

Perhaps there could be some movement of the HMC algorithm description to an appendix?

We have carefully considered this referee's comment, and we believe that only the information strictly pertinent to the specific case study under investigation is presented in detail in the manuscript (e.g., priors, models etc.). More generic non-specific aspects of the method already outlined in Albert et al, 2016, are not covered in this work. Therefore, we would be inclined to keep the description of the HMC algorithm in the main part of the manuscript. We hope to find an agreement with the referee on this point.

**Specific comment 3: Conclusion – could be made stronger to convince the reader of the applicability of the HMC as an inference algorithm for stochastic hydrological models.**

Authors note from addressing previous comments:

Our goal in this work is to present a HMC algorithm that makes Bayesian parameter inference with stochastic models possible, from which hydrology can potentially take great advantages: a sound

calibration of model parameters is essential for making robust probabilistic predictions, which can certainly be useful in planning and policy making. Discussing specific hydrological models or systems is outside the scope of our present work, and will be the focus of further studies.

And:

In the present work, we apply an HMC method with a novel time-scale separation approach (from Albert et al., 2016) for the first time to a real-world hydrological case study, using real time-series of observed rainfall and outflows. Moreover, we also demonstrate for the first time the ability of the algorithm to reconstruct with great accuracy the unknown true average rainfall over the catchment using only prior knowledge and the observed outflow. The reconstructed precipitation is then used to calibrate the hydrological model parameters, which are thus protected against the degrading effect of the possible rainfall data inaccuracy. This considerably reduces the bias in the inferred parameters, thus leading to more realistic models and reliable runoff predictions.

Suggested structure:

Restate goal of paper and scope.

State novelty – first time to reconstruct…

State what was demonstrated and why biases in the inferred parameters were reduced.

State limitations of the study.

State future work.

State implications for hydrology and why broadscale use of the HMC is advantageous.

To include the above salient points.

We have tried to structure the conclusions following the referee's suggestion. We hope that the current version meets the referee's expectations.

**Specific comment 4: Improve writing. Topics are introduced across two paragraphs or more (e.g. in the introduction). Sentences are too long and too complicated throughout.**

Unfortunately, some of the key messages get lost in the unwieldy structure of the paragraphs and sentences. Suggest containing a topic per paragraph and to shorten sentences. Make sure key messages stand out – that they are simple, clear and succinct. The manuscript could be made more impactful by improving the writing.

We have revised the manuscript paying special attention to the style, especially in the introduction and conclusions, according to the referee's suggestion. We hope that now the key messages stand out more clearly.

**Referee #3 – Report #1**

The authors have responded to all the questions and requests I made in the first round of reviews. Therefore, I recommend publication of this very interesting and well-written paper.

At the authors' discretion, I suggest that the paper could be further improved by addressing the following points:

- What is the mean value of the normal distributions used for drawing the momenta, eq. (23)? Is it unity?

The mean value is zero. We have added this detail to the text.

- If the transformation (17)-(19) was non-linear, would the authors have had to set those equations forward in the problem statement, including eq. (11)-(13), i.e., the Jacobian?

Yes, that's correct. Non-linearities in the transformations (17)-(19) would have to be considered in the calculations throughout the manuscript (e.g., Jacobians). However, since this is not essential to understand the key points of our work, we prefer not to discuss it.

- More than resorting to Meta-Dynamics, which is difficult to perform in high-dimensional spaces, shouldn't the author be more interested in parallel tempering? Wouldn't this also help in assessing what the effect of the specific values of the masses would be?

This is an interesting point. We added references to Parallel Tempering in the text.

- I would be curious to know why 6 evaluations of the derivatives of the posterior are required in the molecular dynamics part. Is the derivation of the energy six times more expensive than its evaluation because these are 6 evaluations, or is it actually 6*6=36 times more expensive?

There are 6 evaluations of the derivatives at each call of the algorithm because we have set the number of integration steps P=3, and each integration step requires 2 calculations of the derivatives (as explained in Albert et al., 2016).

The derivation of the energy is 6x6=36 times more expensive than its plain evaluation.

We have added comments to the text to clarify these interesting points.